

# Detection of mixing and precipitation scavenging effects on biomass burning aerosols using total water heavy isotope ratios during ORACLES

Dean Henze[1], David Noone[1,2], Darin Toohey[3]

[1] College of Earth, Ocean, and Atmospheric Sciences, Oregon State University, Corvallis, OR, U.S.A.
  [2] Department of Physics, Waipapa Taumata Rau-The University of Auckland, Auckland, New Zealand
  [3] Department of Atmospheric and Oceanic Sciences, University of Colorado Boulder, Boulder, CO, U.S.A.

*Correspondence to:* Dean Henze (henzede@oregonstate.edu), David Noone (david.noone@auckland.ac.nz)

**Abstract.** The interaction between biomass burning aerosols and clouds remains challenging to accurately determine in part because of difficulties in using direct observations to account for influences of scavenging and dilution separately from sources. The prevalence of mixing versus precipitation processes in biomass burning aerosol (BBA) laden air over the southeast Atlantic is assessed during three intensive observation periods during the NASA ORACLES (ObseRvations of Aerosols above CLouds and their intEractionS) campaign. Air in the lower

free troposphere (FT) and marine boundary layer (MBL) are distinct, and can be treated as separate analyses, although connections are made where relevant. Aircraft in-situ measurements of total water heavy isotope ratios to are used to assess of  air parcel precipitation history. Subsequently, the isotopic fingerprint of precipitation influence is used to identify the prevalence of wet scavenging of black carbon aerosols. In situ observations in the lower FT are combined with satellite and MERRA-2 data into simple analytical models to constrain hydrologic histories of

BBA-laden air originating over Africa and flowing over the southeast Atlantic. We find that even simple models are capable of detecting and constraining the primary processes at play. Regression of the aircraft data onto a simple model of convective detrainment is used to develop a metric of precipitation history. The approach is supported by complementary analysis using the ratio of black carbon to carbon monoxide (BC/CO). The  method is expanded to test the entrainment and precipitation influences on marine boundary layer air. This is more difficult than the lower

FT analysis since signals are more subtle, and limited by imperfect knowledge of the water and isotope ratios of the entrained airmass at cloud-top. Nonetheless, lower cloud condensation nuclei concentrations occur in the sub-cloud layer coincident with isotope ratio evidence of precipitation, indicating aerosol scavenging in the 2016 and 2018 IOPs. For the 2017 IOP, with the highest sub-cloud CCN concentrations, there is no connection between precipitation signals and CCN concentrations, likely indicating the importance of the different geographic sampling

and air mass history in that year. These findings demonstrate the value of leveraging the isotope ratio signals of precipitation history that is distinct from the signature of dilution effects to constrain BC scavenging coefficients in a manner which can guide in model parameterizations, and ultimately lead to improvements in the accuracy of simulated BC concentrations and lifetimes in climate models.



## 1 Introduction

Cloud-aerosol interactions are primary sources of uncertainty in anthropogenic climate forcing (IPCC 2013). Biomass burning (BB) is a key source of aerosols (Bond et al., 2013), and central/southern Africa accounts for almost one third of BB emissions (550 TgC yr$^{-1}$; van der Werf et al., 2010). Black carbon (BC) is a component of African BB with potentially the strongest climate forcing effects. The atmospheric lifetime of BC determines its atmospheric burden and contribution to radiative impacts (Bond et al., 2013; Liu et al., 2020). Despite low hygroscopicity when initially formed, BC can serve as cloud condensation nuclei, particularly after again (Zuberi et al., 2005; Zhang et al., 2008; Tritscher et al., 2011; Twohy et al., 2021). The main removal mechanism of BC is wet scavenging by clouds and precipitation (Latha et al., 2005; Textor et al 2006). Therefore, quantifying processes controlling BC scavenging efficiencies are of importance for climate models.

The BB aerosols (BBA) originating over central/south Africa are routinely transported across the South Atlantic basin (Chand et al., 2009; Zuidema et al., 2016) in the free troposphere (FT) at 2 – 6 km, and may subsequently subside and interact with the large, semi-permanent stratocumulus cloud deck in the Southeast Atlantic. One way aerosols could potentially change cloud cover is by altering cloud lifetimes (Albrecht, 1989; Ackerman et al., 2000; Wood 2007). For example, high aerosol loads may alter cloud droplet size distributions and suppress precipitation, in turn increasing cloud lifetimes. Both the magnitude and sign of the lifetime effect are not sufficiently constrained (Redemann et al., 2021; IPCC 2013).

A component of both BC lifetimes in the atmosphere and cloud lifetime effects is aerosol interaction with precipitation. Therefore, obtaining good observational constraints of the removal of water and aerosols by precipitation relative to the reduction in concentrations due to dilution  is important to constrain coincident behavior of clouds and aerosols. Measurements of the heavy water isotope ratios of water vapor and total water concentrations provide a  basis for a solution to these challenges, because they provide information on the relative importance of air mass mixing, precipitation, and other moisture transport processes that is not easy to determine from conventional thermodynamic variables alone (Galewsky and Hurley, 2010; Noone et al., 2011; Risi et al., 2012; Bailey et al., 2013; Benetti et al. 2015; Steen-Larsen et al. 2015). While isotope ratio information to date has shown promise, the needed vertical profile information on isotopic compositions in the lower troposphere and the marine boundary layer (MBL) are limited. Therefore, an experimental design that integrates isotope ratio measurements paired directly with the complementary thermodynamic and aerosol measures is required for the advantages to be realized.

The analysis presented here utilizes an extensive aircraft in-situ dataset taken during the NASA ORACLES (ObseRvations of Aerosols above CLouds and their intEractionS) campaign (Zuidema et. al., 2016; Redemann et. al., 2021). In addition to in-situ thermodynamic, trace gas, and aerosol variables, the dataset includes in-situ measurements of the isotopic compositions for water D/H. The campaign involved three, month-long sampling periods in the southeast Atlantic (SEA) FT and stratocumulus-topped marine boundary layer (SCMBL), targeted at sampling African BBA plumes and the SEA cloud deck. This study assesses both the FT and SCMBL measurements



to describe the mean atmospheric state and the interaction between distinct atmospheric layers. . In the case of the

FT, , regional aerosol and moisture transport are connected and characterized (e.g. Fig. 1a). The relative importance

of evapotranspiration, dry versus moist convection, and precipitation in controlling isotope signals of FT air

originating over Africa are

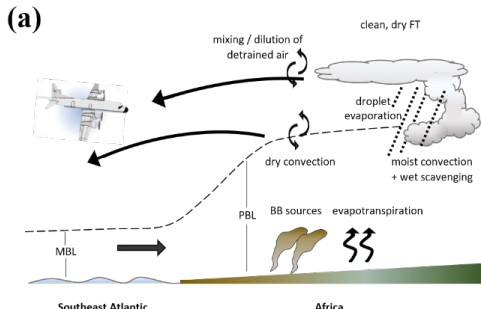

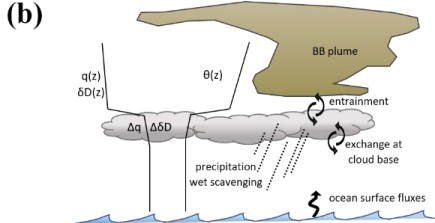


**Figure 1**. **(a) Regional moisture and aerosol transport associated with BBA plumes in the Central African and Southeast Atlantic regions before FT sampling during ORACLES. Processes expected to alter moisture, isotope ratios, and aerosol concentrations are shown. (b) Stratocumulus topped marine topped boundary layer schematic, showing relevant processes to this study. Characteristic potential temperature (θ), total water (q), and HDO/H$_2$O ratio (δD) profiles are included.**


constrained. Hydrologic histories of high versus low BBA loaded air masses are distinguished, and correlation

between isotopic and aerosol is used as evidence that identifies wet scavenging. Next, the occurrence of entrainment

mixing and precipitation in the stratocumulus topped boundary layers is evaluated by utilizing the isotopic

difference between sub-cloud and cloud layers. Correlation between processes diagnosed with the isotope ratios and

variation in sub-cloud aerosol abundance is also assessed. Both the FT and SCMBL analyses leverage expectations

for the relationships between the isotope ratios in relation to water concentration with simple analytical models as a

diagnostic framework. Section 2 covers the ORACLES sampling region, data, and basic isotope theory. Section 3

presents a brief analysis of the sub-cloud as a starting point for contrasting the analysis of the FT analysis of

precipitation histories and scavenging. Section 4 returns to the case of the SCMBL to test the degree to which the





precipitation metrics can be applied to that very different hydrological setting. Section 5 provides final remarks and conclusions.

## 2 Methods

### 2.1 ORACLES sampling region

An extensive overview of the ORACLES project is presented by Redemann et. al., (2021) Aspects of the project and ORACLES measurements relevant to this study are outlined here. In-situ sampling aboard the NASA P-3 Orion aircraft covered the southeast Atlantic MBL and LCLT during periods of high aerosol loading. The high BBA concentrations in the southern hemisphere spring are due to BBA-loaded air in the African PBL being carried out over the SEA by LCLT easterly flow (Garstang et al., 1996; Adebiyi and Zuidema, 2016). This air settles over the SEA cloud deck due to large scale subsidence, where it may then entrain into the MBL. The large-scale subsidence also plays a role in the strong inversion topped MBLs of the region; the MBLs transition to cumulus-coupled SCMBLs toward the equator as sea surface temperatures (SSTs) increase (Wood 2012 and references therein).

Sampling flight tracks are shown in Figure 2. Sampling took place over three intensive observation periods (IOPs): Aug. 31 – Sept. 25, 2016 (14 flights); Aug. 12 – Sept. 2, 2017 (12 flights); and Sept. 27 – Oct 23, 2018 (13 flights). Flights were typically every 2-3 days. Most flights were 7-9 hours in duration and within 7am to 5:30pm local time. Vertical profiles spanned the altitude range 100m to 7km, covering the MBL and the region of the FT where BB plumes were present.

### 2.2 P-3 variables used

Total water mixing ratio ($q$) and heavy isotope ratio $R_D = q_{HDO}/q_{H2O}$ were measured with the Water Isotope System for Precipitation and Entrainment Research (WISPER) instrument package, which centered around two Picarro isotopic gas analyzers (models L-2120fxi and L-2120i). The isotope ratios are reported in delta notation, the deviation of $R_D$ from that of Vienna Standard Mean Ocean Water: $\delta D \equiv (R_D/R_{D,SMOW} - 1)$. A detailed description of those data and calibration methods is given by Henze et al. (2022). Measurements of potential temperature ($\theta$) were computed from measurements of temperature ($T$, Rosemount 102 type non-deiced probe) and static and dynamic (ram) pressures. Refractory black carbon (rBC) and carbon monoxide (CO) are used as indicators of BBA loading in this study. rBC concentration from 53 to 524 nm mass equivalent diameter and adjusted to standard temperature and pressure was measured using a Droplet Measurement Technologies single-particle soot photometer (Stephens et al., 2003; Schwarz et al., 2006). CO concentrations were measured by an ABB–Los Gatos Research $CO/CO_2/H_2O$ analyzer (Liu et al., 2017). Cloud condensation nuclei (CCN) concentrations at 0.3% supersaturation were measured by a Droplet Measurement Technologies CCN-100 continuous-flow streamwise thermal-gradient CCN chamber (Roberts and Nenes, 2005). Sub-cloud CCN measurements were used to estimate the total amount of hygroscopic aerosols present in sampled MBLs. Measurements for $q$, $\delta D$, rBC, and CCN were sampled from a forward-facing



solid diffuser inlet outside the P-3 cabin in front of the wing (McNaughton et al., 2007), operated by the Hawaii
Group for Environmental Aerosol Research and maintained at near isokinetic flow. CO measurements were made
from a rear facing Rosemount inlet probe.

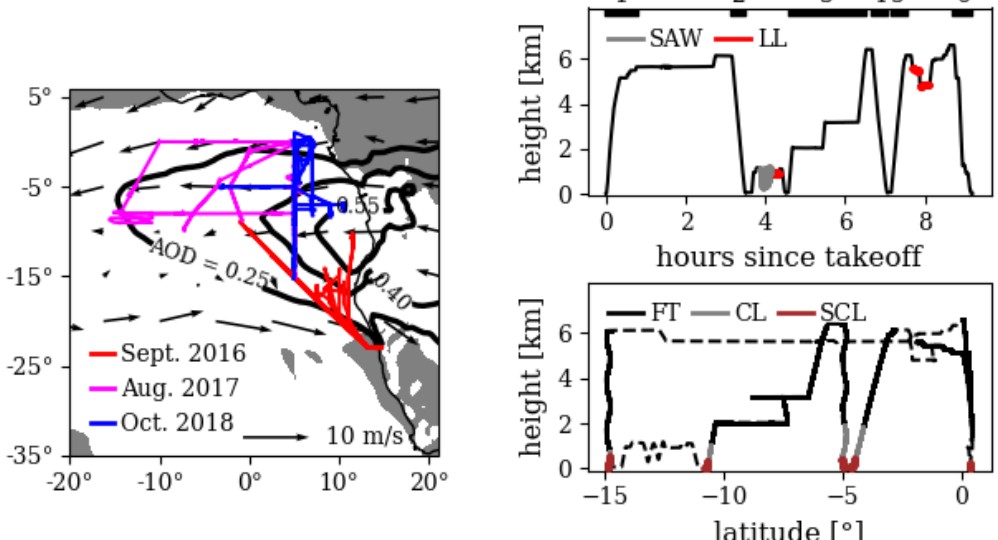

**Figure 2. (Left) ORACLES P-3 Orion flight tracks for the three IOPs. The majority of the 2017 and 2018 IOP flight**
**tracks are co-located geographically (blue north-south track near lon 5W). The Sept. 2016 MERRA monthly mean**
**aerosol optical depth (black contours) and 500 hPa winds (arrows) between latitudes 25°S and 8°N are shown. Shading**
**indicates where the Sept. 2016 MERRA monthly mean 500 hPa vertical velocity is upwards. Aerosol optical depth, 500**
**hPa winds, and 500 hPa vertical velocity are qualitatively similar for the Aug. 2017 and Oct. 2018 observation periods.**
**(Top right) P-3 height vs. time for the flight on Aug. 24, 2017. In-cloud level legs (red) and sawtooth patterns (grey) are**
**highlighted. Black bars with numbers on top indicate manually identified vertical profiles. (Bottom right): P-3 height vs.**
**latitude, with SCL, CL, and FT regions distinguished using the vertical profiles in the top panel and the methods in**
**Section 2.3.**

## 2.3 SCL, CL, and FT identification in vertical profiles

For each flight, time intervals of vertical profiles were flagged manually. To maximize the amount of data used in
this study, profiles were liberally identified as any flight segment with a progressively upward or downward
trajectory from near the surface (100 – 200 m) to at least 2500 m (e.g. Fig. 2). For each profile, vertical profiles of $T$,
$\theta$, and relative humidity ($RH$) averaged into 50 m vertical bins were used to identify the well-mixed layer top and
MBL capping inversion bottom as described below. Data below the well-mixed layer top are flagged as SCL, data
between mixed layer top and inversion bottom are flagged as cloud layers (CL), and data above inversion bottom are
flagged as FT. Using vertical profiles of $T$, $\theta$, and $RH$ was chosen over liquid water content as a flag for the CL since



it would also capture profiles through CLs with broken clouds, where by chance the P-3 avoided clouds during vertical ascent or descent.

Mixed layer tops were roughly taken to be the height at which potential temperature increased by 0.2 K from its 100 – 300 m mean. A potentially more rigorous identification of cloud-base using lifting condensation levels was not used since data below 100 m was not always available. The MBLs in the study region were almost always characterized by a capping temperature inversion and a peak in *RH* at their top. Inversion bottoms were identified as the highest altitude below 2.5 km where a temperature minimum coincided with an RH maximum to within 150 m. The temperature minimum relative peak had to be at least 1 K and occur within a vertical region of 400 m or less. The RH absolute peak had to be greater than 80% and occur within a vertical region of 300 m or less. Figure 2 shows an example flight using the above methods to identify SCL, CL, and FT. Using this method, the majority of data are flagged as one of the three regions and are able to be used for analysis.

**2.4 Cloud layer modules**

Most flights involved modules to directly target the CL. These included: (1) level legs through the cloud layer at a single altitude (LLs), typically lasting 5 – 15 minutes, (2) sawtooth patterns through cloud layer (SAWs), also typically 5 – 15 minutes, where individual saw teeth spanned the full vertical extent of the cloud layer (Fig. 2). Both LL and SAW modules were previously flagged (Redemann et al., 2021) as used in Diamond et al., 2018. Where available, these modules are preferred over the CL flagging in Section 2.3. For LLs, data were averaged into 3-minute blocks. At an aircraft horizontal speed of 130 m s$^{-1}$, each block represents almost 25 km of horizontal distance, equivalent to the horizontal grid spacing of higher resolution GCM simulations. SAW patterns included periodic portions above and below the CL. To ensure only the CL portioned are used, times where RH drops below 99% are discarded. The remaining data are averaged into 3-minute blocks. Only LLs and SAWs below 2.5 km are used.

**2.5 Identifying distinct FT airmasses**

Distinct FT airmasses in each vertical profile were identified using a pseudo-conservative variable approach to highlight the differing hydrologic histories rather than the subsequent mixtures between them. Simple two-endmember mixing processes are found as straight lines on the diagram while curved segments signal non-conservative processes such as precipitation, or more complex mixing environments (such as multiple sources or time evolving end members). Diagrams of q vs equivalent potential temperature ($\theta_e$) have been used before to identify regions of mixing in atmospheric profiles as well as the mixing endmembers (Paluch, 1979; Betts and Albrecht, 1987). Here, instead of $\theta_e$, the quantity $q\delta D = q_{HDO}/R_{D,SMOW} - q$ is utilized (since the mixing ratios $q_{HDO}$ and q are both conserved in mixing processes, so is the linear combination q $\delta$D).

P-3 vertical profiles as identified in Section 2.3 were averaged into 25m altitude bins and their trajectories in *q* vs *q$\delta$D* space were visually broken into piecewise segments. Figure 3 shows an example of segment identification, with



associated profiles of thermodynamic quantities and the BBA indicator rBC. The diagram of $q$ vs $q\delta D$ shows a series of piecewise segments, where the corners/kinks are taken as the independent endmembers (open circles). The vertical profiles show that these air masses correspond to minima/maxima in the profiles of $q$, $\delta D$, and rBC, or are near kinks in the $\theta$ profile, supporting the validity of this method. Although some of these airmasses reside in the MBL, only the FT airmasses were used.

**2.6 Analysis of airmass hydrologic histories using humidity-isotope pairs**

The $\delta D$ of an air mass relative to its water vapor mixing ratio, q, will assume different relationships depending on the hydrologic processes the air mass experiences. PDFs in (q, $\delta D$) space can be compared to theoretical curves (e.g. Figure 3) to constrain these processes. A survey discussion on the theory is presented in Noone (2012) (note that "$q$" in Noone 2012 denotes water vapor while here denotes total water). Some key components are presented here to

facilitate discussion below.

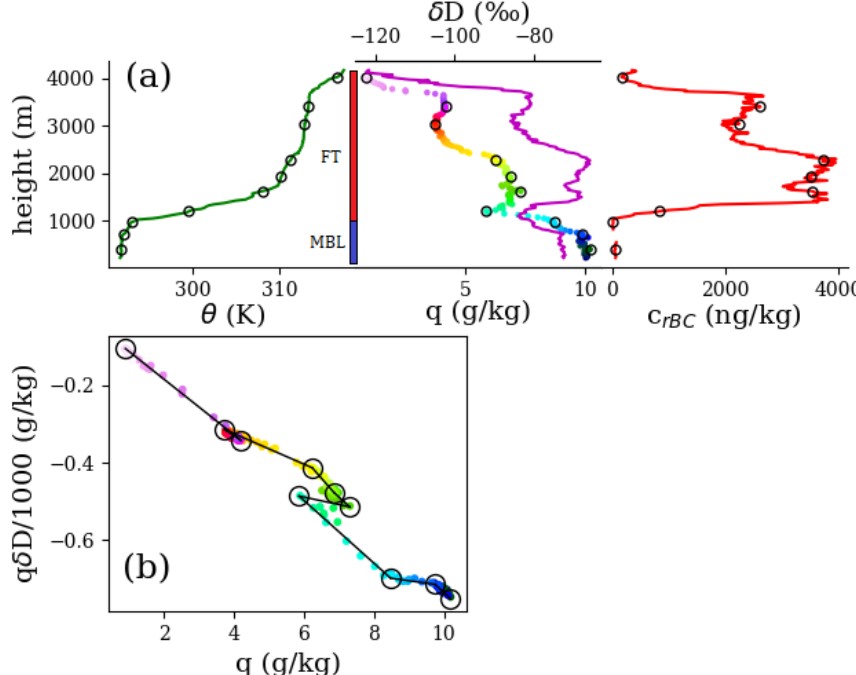

**Figure 3**. (a) One of the P-3 vertical profiles on the Sept. 4, 2016 flight, averaged into 25m bins. Profiles of in-situ potential temperature (green), total water mixing ratio (multi-color), $\delta D$ (magenta), and refractory black carbon (red). (b) q vs. q$\delta D$ for the profile, with the same altitude-coloring as the total water profile. Open circles in (a) and (b) mark distinct air masses as outlined in Sect. 3.3.5, identified visually from (b).



Figure 4 shows the ($q$, $\delta D$) value of an asymptotic endmembers for dry free tropospheric (FT) air and pure surface evaporation from land and ocean sources. Land surface evapotranspiration (ET) may have higher $\delta D$ values than ocean evaporation since transpiration does not fractionate heavy isotopes. The mixing trajectories (black, grey) show the $q$, $\delta D$ relationship for air where moisture from the respective evaporation source and is diluted with FT air to varying degrees. In general, air that has not experienced condensation processes may be modelled as a mixture of two or all of these endmembers. The blue circle in Figure 3 shows an example MBL placement on the mixing line assuming 80% relative humidity with respect to a sea surface temperature of 293K. Further entrainment of FT air into the marine boundary layer (or conversely detrainment of MBL air into the FT) would continue along the mixing trajectory. On the other hand, precipitation preferentially removes heavy isotopes, resulting in a steeper trajectory. The black-dashed curve in Figure 4 shows the extreme case of 100% condensate removal (Rayleigh process).

Air that experiences either condensate removal at less than 100% efficiency or a combination of precipitation and mixing will fall between the grey solid and black dashed curves. If we assume a simple convective plume model with no entrainment, a convecting airmass undergoing a Rayleigh process would first follow the dashed curve from the blue circle, but then divert along the 'convective detrainment' mixing curve (solid line) upon reaching a detrainment level, when the air mass begins to dilute with FT air. In this study, detrainment heights and temperatures are estimated along the Rayleigh trajectory as an estimate of detrained water, and is not substantially different from a partial removal model (e.g., Merlivat and Jouzel, 1979, Noone 2012) at high water vapor mixing ratios. The algorithm used has the following features (1) Values of surface $q$, $T$ and elevation are estimated from MERRA and used as initial conditions. (2) A temperature profile is computed from these surface values, assuming a dry adiabatic lapse rate up to a lifting condensation level (LCL) and a moist lapse rate above the LCL. (3) For each q value along the Rayleigh trajectory, a dewpoint is calculated as the detrainment temperature and the height at which the dew point intersects the temperature profile is the detrainment height. Figure 4 shows a characteristic case with the dew point and height from this computation for the example MBL undergoing convection and detraining at $q = 6$ g kg$^{-1}$.



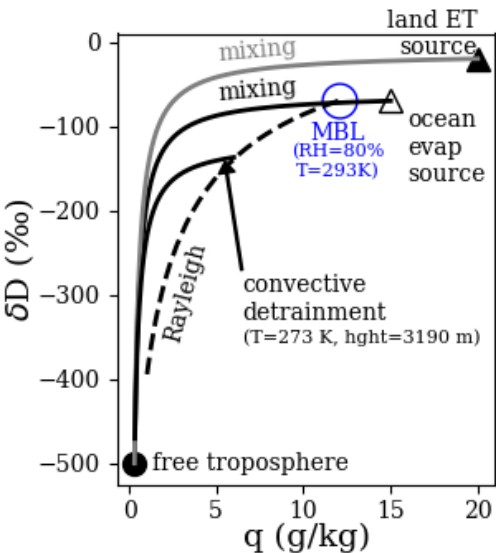

**Figure 4**. **Theoretical trajectories in (*q*, *δD*) space. Equations to generate curves were taken from Noone 2012.**

## 3 Results

### 3.1 Sub-cloud layer humidity and δD variance

Figure 5 shows of SCL data averaged into 60 s blocks, where the SCL was identified as in Sect. 3.3.3. Data below 150 m are colored by SST. SST was determined by collocating NOAA COBE monthly mean SST satellite data with the position of the aircraft. Previous studies have been devoted to a detailed understanding of the isotopic composition of the SCL, typically using either land or ship-based measurements (e.g. Benetti et al., 2018; Feng et al., 2019). Rather than testing those analyses in detail, the ORACLES SCL data are shown here briefly for context as a basis for the novel aspects of the aircraft profile dataset. Nonetheless, Figure 5 includes $\delta D$ predictions using the simple model of Merlivat and Jouzel, 1979 (MJ79). The MJ79 model – which assumes SCL moisture comes solely from ocean evaporation - was not originally intended for regional scale and sub-daily predictions and its limitations in doing so have been investigated elsewhere (e.g. Jouzel and Koster, 1996; Kurita, 2013). The MJ79 model reduces SCL $\delta D$ to a simple function of SST and relative humidity with respect to SST (*RHS*), and provides a useful reference point. For each 60 s average, the MJ79 formula was used for ocean $\delta D$ values in the range of 0 ‰ to 6 ‰ for the 2017 and 2018 IOPs, and in the range 0 ‰ to 10 ‰ for 2016. The range was used to account for uncertainty in the ocean isotopic values which were not sampled during ORACLES. The upper bounds were taken by inspection of Fig. 4 in Benetti et al. (2017).





The right column of Fig. 5 provides a metric of variability in the SCL. The histograms show the difference between means of the lower and upper most sampled 100 m of the SCL for each vertical profile in Section 2.3. The standard vertical difference is about 1 ‰ for 2016 and 2 ‰ for the other two IOPs. Horizontal variability is also explored. Horizontal aircraft legs below 500 m and lasting at least 3 minutes were averaged into 10 s blocks and then the

standard deviation of the entire leg was computed. The median horizontal standard deviations are roughly the same as the standard vertical variabilities for 2016 and 2017. For 2018, the median horizontal variability is 1.25 ‰, slightly less than the 2 ‰ vertical variability.

### 3.2 FT rainfall and evapotranspiration histories from satellite data

One objective of this study is to build a characterization of rainfall and evapotranspiration histories for FT air

sampled during ORACLES using the humidity and isotopic data. For comparison, a characterization using airmass back-trajectories and satellite products of monthly mean rainfall and evapotranspiration is also developed. Figure 6 shows six-day back-trajectories computed with HYSPLIT (Draxler and Hess, 1997, Warner, 2018) via the PySplit Python module (Warner, 2018) for FT airmasses sampled by the P-3 (HYSPLIT runs used Global Data Assimilation System output on pressure levels at 1˚ x 1˚ resolution and 3-hour frequency). The trajectories are separated by

sampling period and BBA loading histories inferred from their in-situ CO concentrations. Airmasses where the in-situ CO concentrations were higher vs. lower than the FT median for that sampling period are referred to hereafter as high-BBA vs. low-BBA are, respectively. It is noteworthy that a large fraction of the "low-BBA" can have CO concentrations double that of the SCL (Fig. 6c). For 2016, high-BBA originates from the highest latitudes out of the three years. For 2017, both high-BBA and low-BBA air originate further north and their latitudes tend to overlap.

For 2018, the low-BBA trajectories are on average displaced 3˚ north of the high-BBA trajectories, toward regions of higher rainfall. The high-BBA air originates from a region similar to that of 2016 but whereas the 2016 high-BBA air flows west off the continent, the 2018 high-BBA air tends to flow north-west. The low-BBA air in 2016 it is distinct as it often originated from higher latitudes of the Atlantic Ocean.





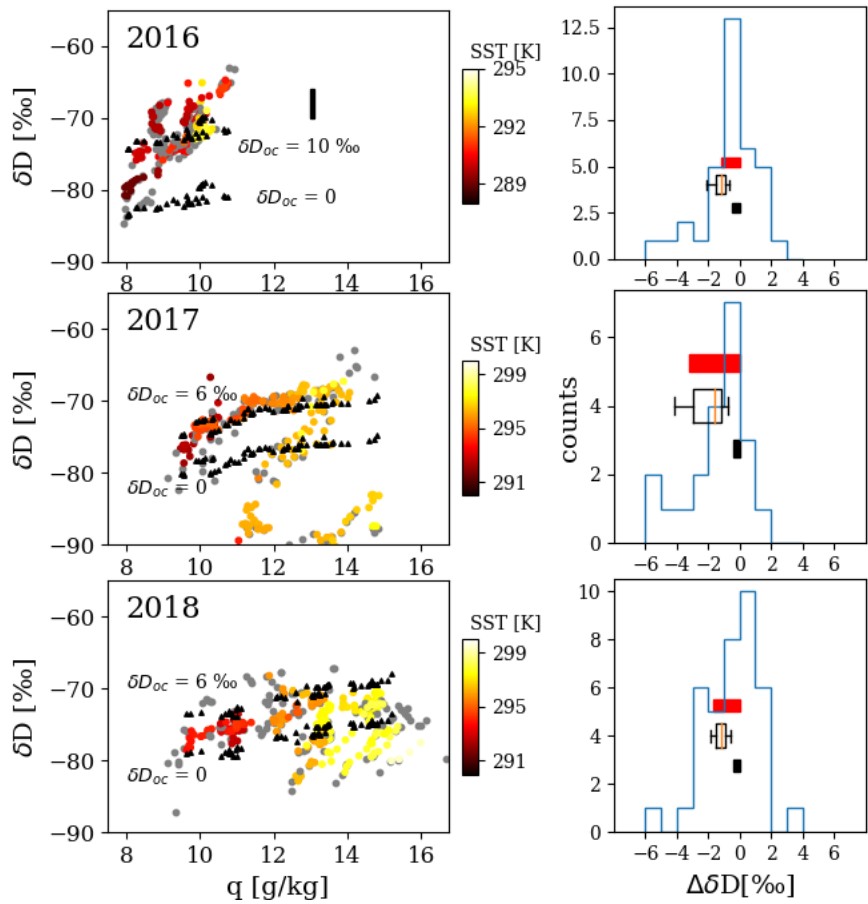

**Figure 5**. **(Left column) ($q$, $\delta D$) data in the SCL for 2016 (top), 2017 (middle), and 2018 (bottom), averaged into 60 s blocks. Measurements below 150 m height are colored by NOAA COBE monthly mean satellite SST. Black triangles show predictions using the Merlivat and Jouzel (1979) closure assumption. The vertical black bar displayed for 2016 is the instrument standard error on an absolute scale. (Right column) Histograms of the difference in $\delta D$ between the top and bottom most sampled 100 m of the SCL for each vertical profile. Red bars are the standard deviations of the histograms. Box plots are the $\delta D$ standard deviations of all P-3 horizontal legs below 500 m height and lasting at least 3 min averaged to 0.1 Hz (n = 23, 22, 25 for 2016, 17, 18 IOPs). Black bars are 0.1 Hz instrument precisions. All standard deviations are multiplied by -1 for visuals.**



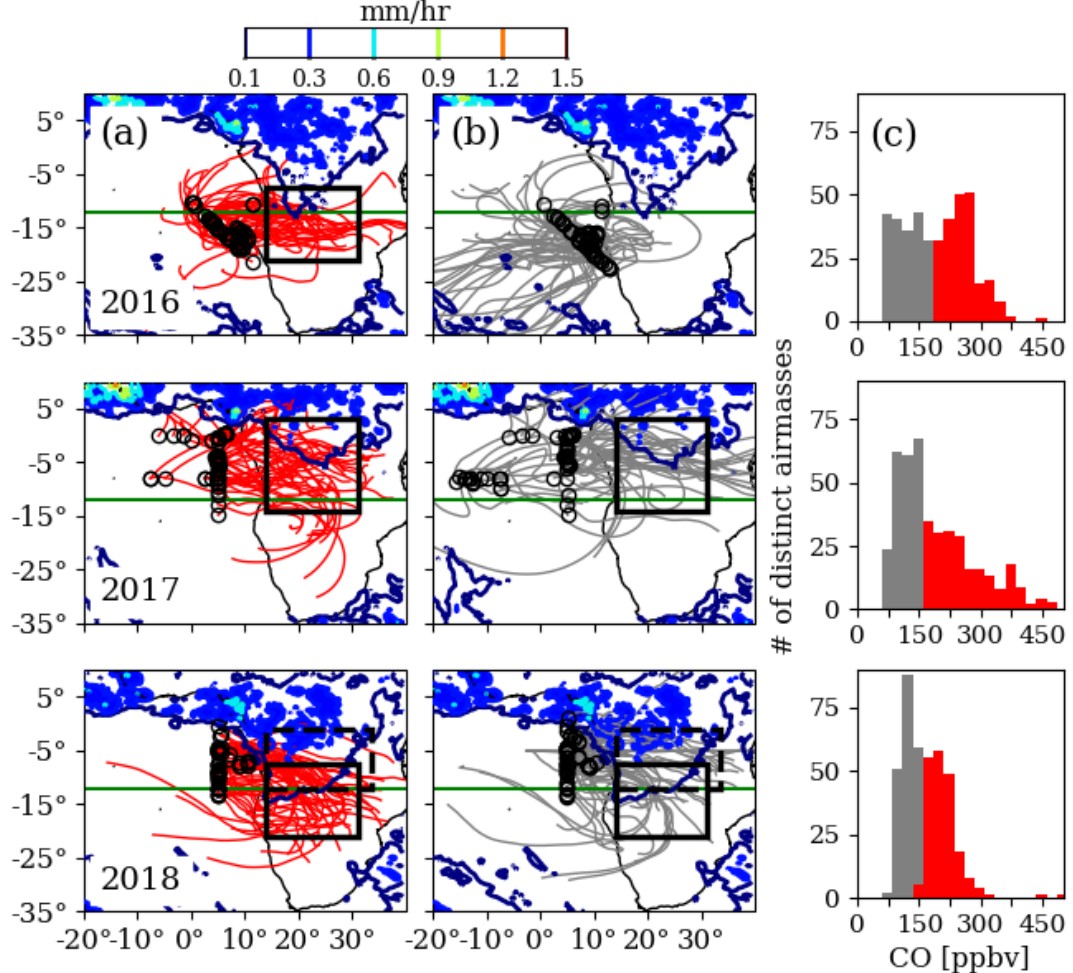

**Figure 6**. **HYSPLIT back-trajectories for distinct FT airmasses (identified using the methods in section 2.5) with in-situ CO greater than the median (a) vs. less than the median (b). CO histograms are also shown (c). Back-trajectories are shown for the Sept. 2016 (top row), Aug. 2017 (middle) and Oct. 2018 (bottom) IOPs. Black open circles mark trajectory end points - the locations of P-3 in-situ sampling. Overlain are NASA GPM satellite monthly mean rainfall maps for Sept. 2016, Aug. 2017, and Oct. 2018. Boxed regions mark areas over which mean MERRA and CAM values were taken for calculations discussed in the main text and outlined in Table 3.3. The green line at 12˚S is a visual reference. Median FT CO values for the 2016, 2017 and 2018 sampling periods were 200, 162, and 157 ppbv.**

Across sampling periods there are distinct differences in the back-trajectory collocation with rainfall (Fig. 6). Fig. 7 shows that in 2016, high-BBA trajectories show the lowest monthly-mean rainfall. The frequency along 2016 trajectories where satellite GPM monthly mean precipitation rates are greater than 0.25 mm hr$^{-1}$ is an order of magnitude less than that for the other two years. For 2017, there is no significant difference in the PDFs for high-




BBA vs low-BBA, while in 2018 the low-BBA trajectories are 2-3x more likely than high-BBA to pass over regions
with precipitation > 0.25 mm hr$^{-1}$.


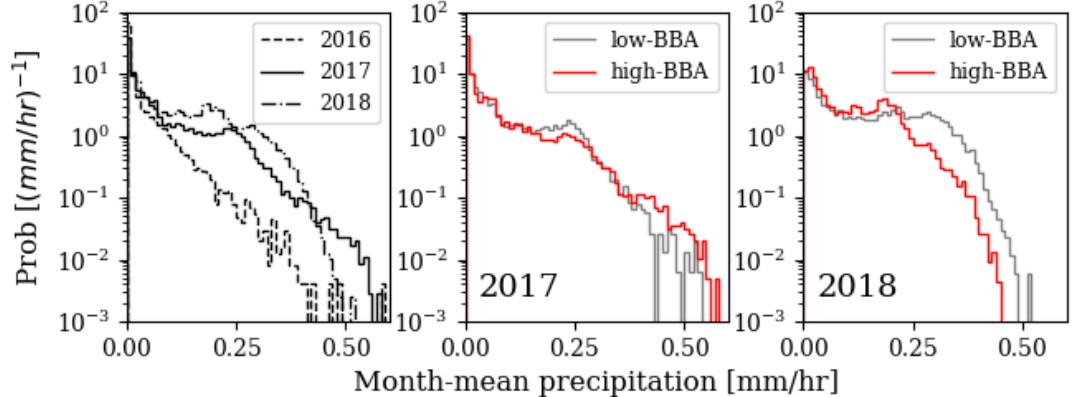

**Figure 7**. Probability distributions of GPM monthly mean rainfall along FT back-trajectories in Fig. 6. Only the portions
of the trajectories over the African continent were used, defined roughly as any portion of the trajectory east of 10˚E.
Distributions between IOPs are compared (left), and between high vs. low BBA air for the 2017 IOP (middle) and 2018
IOP (right).

**Table 1**. Estimated ET contribution to African PBL moisture for each ORACLES sampling period, using a back of the
envelope calculation (Appendix A).

| Fig. 6 Region | MOD16A2 mean ET (kg m$^{-2}$ s$^{-1}$) | Parameter Values | Δt (days) | % ET contribution |
|---|---|---|---|---|
| 2016 solid box | 6·10$^{-6}$ (± 6·10$^{-6}$) | $h$ = 700 m, $q$ = 9 g kg$^{-1}$, $\rho$ = 1.2 kg m$^{-3}$ | 1 | 6% (±4%) |
| | | | 2 | 12% (±7%) |
| | | | 3 | 17% (±10%) |
| 2017 solid box | 3·10$^{-5}$ (±1.2·10$^{-5}$) | $h$ = 1000 m, $q$ = 12 g kg$^{-1}$, $\rho$ = 1.2 kg m$^{-3}$ | 1 | 14% (±5%) |
| | | | 2 | 25% (±7%) |
| | | | 3 | 33% (±9%) |
| 2018 solid box | 1·10$^{-5}$ (±1·10$^{-5}$) | $h$ = 1000m, $q$ = 12 g kg$^{-1}$, $\rho$ = 1.2 kg m$^{-3}$ | 1 | 5% (±5%) |
| | | | 2 | 10% (±7%) |
| | | | 3 | 15% (±9%) |
| 2018 dashed box | 3.5·10$^{-5}$ (± 1.1·10$^{-5}$) | $h$ = 1000m, $q$ = 12 g kg$^{-1}$, $\rho$ = 1.2 kg m$^{-3}$ | 1 | 16% (±5%) |
| | | | 2 | 28% (±7%) |
| | | | 3 | 37% (±8%) |



In addition to precipitation, there is variation in mean evapotranspiration (ET) in the regions from which the
trajectories originate. Because ET contribution can alter the isotopic content of an airmass, its expected fraction of
moisture contribution for each sampling period was estimated using a back of the envelop calculation outlined in
Appendix A.1 and the MODIS/TERRA Version 6 Evapotranspiration/Latent Heat Flux product (MOD16A2), taking
the mean and standard deviation over the boxed regions of Fig. 6. The results are shown in Table 3.1 show expected
ET contributions of roughly 5-15% at higher latitudes and 15-30% for central Africa.

### 3.3 FT rainfall and evapotranspiration histories from in-situ humidity and isotope ratio analysis

The grouping scheme in Section 3.2 (by IOP and BBA loading history) is used to categorize the in-situ data to
determine if similar distinctions in their precipitation and evapotranspiration histories are derived using a $q$-$\delta D$ space
analysis (Figures 8-10). The asymptotic endmembers and theoretical curves reviewed in Section 2.6 are estimated
and plotted in Figures 8, 9, and 10 for each sampling period. The symbols/curves for each quantity/process match
those used in Fig. 4. Table 3.2 summarizes their computation. Some of the endmembers require estimations of near
surface values over either the ocean or African continent, as detailed in Table 3.2. Near-surface temperatures and
water mixing ratios were estimated using monthly mean 10m values from the Modern-Era Retrospective analysis for
Research and Applications, Version 2 (MERRA-2). The $\delta D$ of near-surface air over Africa is taken from output of
the isotopic version of the Community Atmospheric Model simulations (isoCAM5, Nusbaumer et al., 2017) which
were run for the ORACLES time period (2016-2018) and nudged with the MERRA-2 winds in a configuration very
similar to that described by Fiorella et al., (2022). FT air masses are split into high-BBA (red) vs. low-BBA (grey)
as in Section 3.2. Data for high-BBA vs. low-BBA air in figures 8-10 are plotted as joint probability distributions
(PDFs) of $q$, $\delta D$. PDFs were derived using a gaussian kernel density estimation. The kernel bandwidth was
estimated using Silverman's rule of thumb with a fudge factor of ¾ to account for over-smoothing. SCL data are
plotted in blue after 60 s averaging. Horizontally, the SCL data on average fall between 65-80 % $RH$ with respect to
mean SST, which is reasonable.

The blue mixing curves in figures 8-10 use a combination of ORACLES in-situ and MERRA datasets along with the
evapotranspiration estimations to provide more observationally constrained moist mixing endmembers. For the 2016
IOP where ET is thought to be minimal, the mixing curve simply starts at the mean SCL $q$, $\delta D$ values. For the 2017
IOP, ET contribution is estimated as 25 %. The open box in Fig. 9 represents a 75 % / 25 % mixture of mean
ORACLES SCL and a $q$, $\delta D$ point taken from mean 10 m values derived from MERRA and isoCAM5 simulations
10 m over the boxed region in Fig. 6 for 2017. It is taken as the moist endmember for the blue curve. The dashed
boxed in Fig. 10 for the 2018 IOP is an analogous 75 % / 25 % mixture for the dashed box in Fig. 6. The solid box is
provided mostly for reference and is the point along the blue curve where q reaches its MERRA 10 m value over the
solid 2018 box.



**Table 2. Methods for computing theoretical asymptotic endmembers and curves for figures 8, 9, and 10.**

| Endmember/ Process | Symbol/ line | Calculation |
|---|---|---|
| Ocean evaporation source | open triangle | Estimated as air in thermodynamic/isotopic equilibrium with the mean SST over each sampling period/region. SSTs were taken from the NOAA COBE monthly mean product. Isotopic equilibrium was computed using e.g. Noone 2012 equation 6, where equilibrium fractionation factors were computed from formulas in Horita and Wesolowski, 1994. $\delta$D of the ocean water was taken as 3 ‰. |
| Continental ET source | filled triangle (runs off figures) | $q$ was taken as saturation with respect to mean MERRA-2 surface skin temperature in the boxed region of Fig. 9 for the respective sampling period. $\delta$D was taken as the mean $\delta$D surface flux from isoCAM over respective boxed regions. |
| Dry FT air | filled circle | Mean of all ORACLES WISPER in-situ data between 0.25-0.5 g kg$^{-1}$. Whiskers are standard deviations. |
| Mixing of FT air with ocean or land source | solid black and grey curves | Computed with Noone 2012, equation 23. Only a portion of the curves are shown in Figures 8 – 10. |
| Rayleigh (100% condensate removal) | dashed black curve | Rayleigh model (e.g. Craig et al. 1963; Dansgaard 1964). Starting point for each Rayleigh model is given in the respective figure caption. |
| $T_{sat}$ & height along Rayleigh curve | arrows with annotations | As described in the last paragraph of Section 2.6. Near surface T, q, and z for estimating the LCL were taken as mean MERRA-2 10 m temperature, 2 m humidity, and surface elevation in the boxed regions of Fig. 6. |

While the majority of 2016 high-BBA data can be described by dilution of high humidity continental air with clean FT air (the blue mixing curve), the 2017 data fall well below this curve and require more analysis. One possibility is convective detrainment. Assuming an ascending parcel model following a Rayleigh process, $q$ and $\delta$D would fall

monotonically with height as water in the parcel condensed and precipitated out. Detrained air at a specific height would have a characteristic $\delta$D and subsequent mixing with ambient dry air would alter $\delta$D by less than 20 ‰ until the air reached about 2 g kg$^{-1}$ humidity (e.g. the 'detrainment' curves in Fig. 9). For the 2017 IOP, a correlation of -0.67 is found between $\delta$D and altitude, with $\delta$D decreasing at a mean rate of -35 ‰ km$^{-1}$. In comparison, 0.05 correlation is found for the 2016 IOP despite similar sampling altitudes. Further, using the method outlined in

Section 2.6, detrainment heights and temperatures from the Rayleigh trajectory are estimated. The best fit lines (r of 0.68 for each) have slopes of 0.54 (Fig. 9).





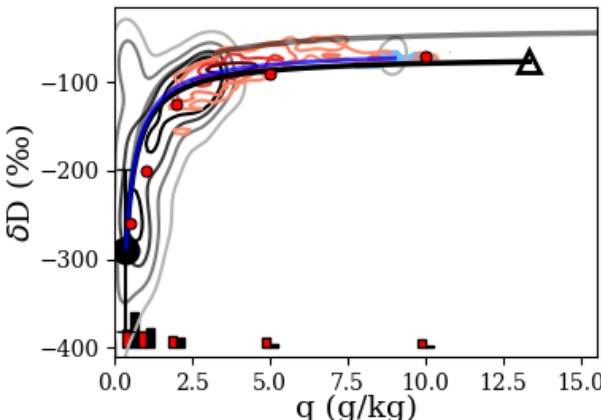

**Figure 8**. (*q, δD*) PDFs for distinct FT airmasses identified from vertical profiles during the Sept. 2016 sampling period. The high-BBA data are shown in red (n=206) and low-BBA data (n=209) in grey. PDFs are expressed as cumulative probability distributions - the contours enclose 25, 50, 75, and 90 % of the data. The blue shaded region encloses all SCL data, averaged into 30 s bins (n=203). The open triangle, filled circle, solid black curve, and solid grey curve are analogous to those in Fig. 4 and are computed as outlined in Table 3.2. The blue curve is a mixing curve with moist endmember centered on the SCL data. Precisions (black bars) and accuracies (red bars) are given near the bottom, where each set of bars corresponds to the small red circle vertically above it.

For the 2018 IOP, the high-BBA air lobe falls along a mixing line with evidence of ET contribution (Fig. 10a), but the PDF extends downward and is bound from below by a detrainment curve at 3600 m. The main low-BBA PDF lobe falls near a Rayleigh path, even extending below. The majority of this signal is due to air sampled below 3000 m. If these data are removed (Fig. 10b) the majority of the signal either falls directly along a Rayleigh path, or in a (q, δD) region indicative of convective detrainment as in 2017. An estimation of altitude and temperature were performed using the Rayleigh model separately for the high vs. low BBA data. Correlations of 0.5 with observed values were found for each group. The low-BBA air has best fit slopes of 0.48 and 0.51 for altitude and temperature. The high-BBA air has best fit slopes of 0.71 and 0.86. δD decreases with altitude by -19 ‰ km$^{-1}$ and -12 ‰ km$^{-1}$ for the low and high BBA groups respectively.

### 3.4 Evidence of BC scavenging from combined BC/CO and (*q, δD*) measurements

Black carbon can serve as CCN and be permanently removed via precipitation. Carbon monoxide, on the other hand, has a low solubility in water and therefore is not altered by clouds. Consequently, scavenging of BC by precipitation should cause the BC/CO ratio to decrease (e.g. Liu et al., 2020). The FT (*q, δD*) diagrams for the 2017 and 2018 IOPs show that airmasses sampled had a variety of integrated precipitation amounts, providing an approach to test for evidence of BC scavenging (2016 was not used because there is minimal occurrence of precipitation). Unlike for the previous section, where distinct airmasses were isolated in order to highlight the primary hydrologic processes occurring, all data that likely originate from the African PBL are used here. The motivation is to diagnose the





influence of scavenging on rBC concentrations for an arbitrary parcel of air. FT data were averaged into 60-s blocks and data with CO less than 100 ppbv were discarded, as they reflect air of marine origin.

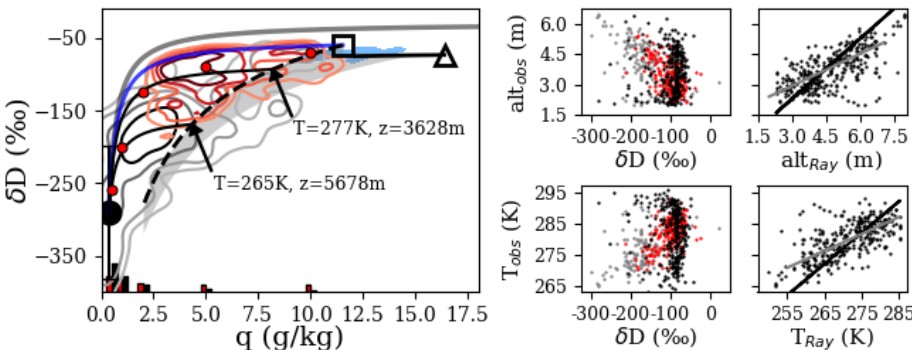

**Figure 9**. (left) Similar to Fig. 8 but for the ORACLES 2017 sampling period. Red PDF (n = 218), grey PDF (n = 217), blue shaded region (n = 902), open triangle with black line segment, grey, and blue curves are as in Fig. 8. The open square represents mean MBL adjusted for estimated ET contribution as described in Sect. 3.4.3. The blue solid curve is a mixing model. Thin black curves (solid) from the Rayleigh curve (dashed) are detrainment mixing models as described in Section 2.6. The light grey shaded region bound Rayleigh processes for all measured SCL initial values. (Center column) $\delta D$ vs height and temperature for 2017 (red, grey) and 2016 (black) data. (Right column) observed heights and temperatures vs. those estimated using the Rayleigh model as detailed in Section 2.6, along with best fit (grey) and 1-1 (black) lines.


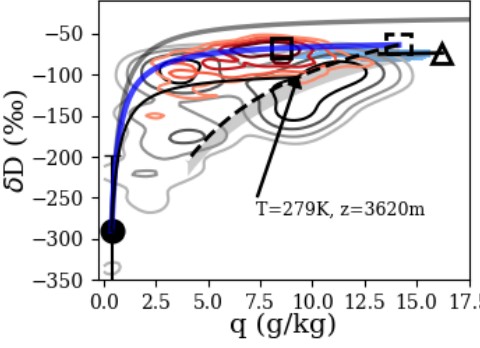

**Figure 10**. Similar to Fig. 9 but for the ORACLES 2018 sampling period. All symbols and curves are analogous to those in Fig. 9. n = 200, 200, and 780 for the red PDF, grey PDF, and blue shaded regions respectively. The open square near the red PDF is the point along the blue curve where $q$ is equal to its mean MERRA 10m value over the solid boxed region of Fig. 6.


Figure 11 shows clear gradients in BC/CO for $q$-$\delta D$ space. If one uses the simple detraining convective plume model (outlined at the end of Section 2.6) to interpret the joint $q$-$\delta D$ data, then the BC/CO decreases with increased





precipitation amount, which would be expected for scavenging. Because the detrainment mixing curve has little $\delta D$
variability for $q > 2$ g kg$^{-1}$, the boxed regions depicted in Fig. 11 correspond roughly to relatively low, moderate, and
high precipitation amounts. PDFs of BC/CO for these three regions (Fig. 12a,b) separate with statistical significance
(at the $\alpha$=0.05 using a t-test) for both the 2017 and 2018 IOPs. For the 2017 IOP the PDFs means are 9.1, 5.9, and
3.9 ng kg$^{-1}$ ppbv$^{-1}$. For 2018, the means are 5.1, 3.4, and 1.5 ng$^{-1}$ kg$^{-1}$ ppbv. As a control, the data separated by
relatively low, moderate and high values of $q$ (bounds of (2,5), (5,8), and (8,11) g kg$^{-1}$ respectively) show no
statistically significant separation.

The change in $q$ due to precipitation, $\Delta q_{precip}$, is computed for each data point in Fig. 11 using an approximation to
the convective plume model. For $q > 2$ g kg$^{-1}$, we take the observed $\delta D$ to be equal to that of the plume when it
detrained, so that $\Delta q_{precip}$ can be approximated with the Rayleigh model as.

$$\Delta q_{precip} \sim \left| q_0 - q_{Ray}^{-1}(\delta D_{obs}) \right|, \tag{1}$$

where $q_0$ is the estimated initial $q$ of the plume before precipitation, $q^{-1}_{Ray}$, is the Rayleigh model inverted to predict
$q$ as a function of $\delta D$, and $\delta D_{obs}$ is the observed value. This is difference is intuitively similar to the "rainout
fraction", $F = q/q_0$), that appears in Rayleigh calculation. The correlation between $\Delta q_{precip}$ and BC/CO is 0.71 for
2017 and 0.56 for 2018 (Fig. 12c). In comparison, the correlation between $q$ and BC/CO is 0.21 for 2017 and -0.14
for 2018, and distinctly inferior. This result suggests a quantifiable path for deriving BC scavenging coefficients
using isotopic data that is unavailable from measures of water concentrations alone.

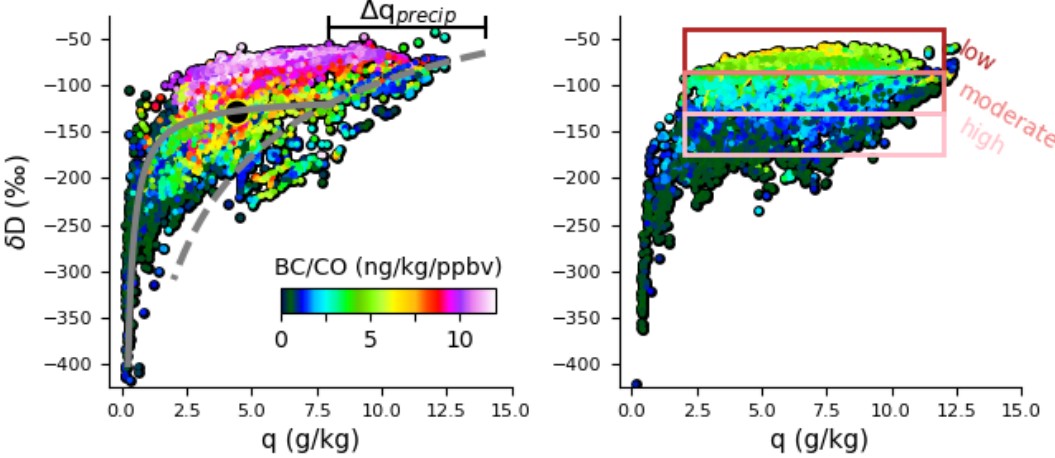

**Figure 11.** *$q$-$\delta D$ diagrams for 2017 (left) and 2018 (right) IOPs, colored by rBC/CO. Grey lines show schematically the
regression of the detraining convective plume model onto a sample data point (black filled circle), and the estimated
change in $q$ due to precipitation. Rectangles (shades of red) show the approximation of this model applied in Figure 12 to
group a subset of the data into low, moderate and high precipitation amounts.*

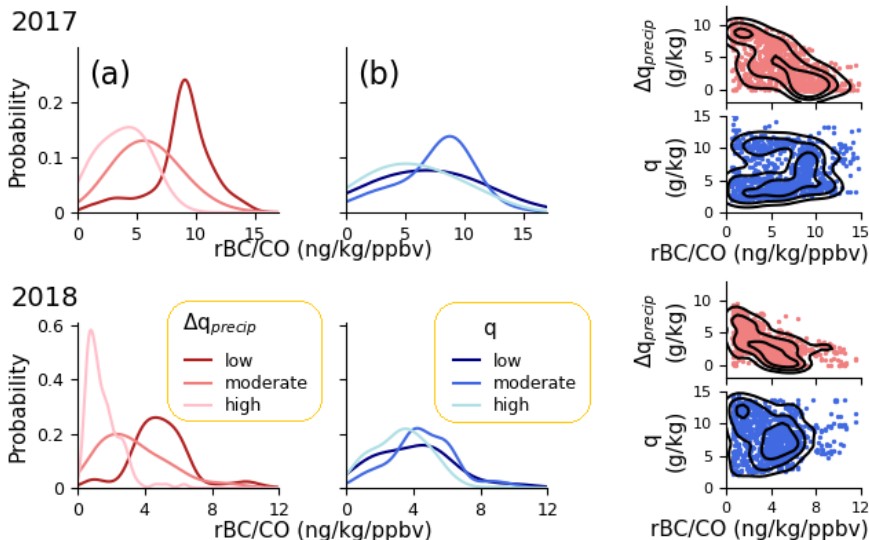

**Figure 12**. Evidence for BC scavenging in FT BBA air using (*q, δD*) data. (a): rBC/CO PDFs for low, moderate, and high
*Δq$_{precip}$* as defined in Fig. 11, for the 2017 (top row) and 2018 (bottom) IOPs. (b): rBC/CO PDFs for Fig. 11 data grouped
by low, moderate, and high values of *q* as detailed in the main text. (c): scatter plots of rBC/CO versus Δq$_{precip}$ and *q*, with
PDF contours overlain.

### 3.5 Entrainment mixing and precipitation in the SCMBL

Figure 5 shows the range of SCL (*q, δD*) states captured during ORACLES. This variability is small in comparison
to the isotopic signal of hydrologic histories of FT air masses, e.g. the *δD* of the moist endmember is nearly constant
given the dynamic range of values higher up. However, it was found that for MBLs sampled during ORACLES, the
differences between the SCL and CL are small enough to be comparable to or less than SCL variability over the
observation period and region. This range makes it more challenging to attribute variations in CL (*q, δD*) to specific
CL processes (e.g. entrainment vs. precipitation) on an absolute scale given the measurement uncertainties.
Therefore, CL (*q, δD*) data are plotted as deviations (*Δq, ΔδD*) from the mean values of the nearby sampled SCL, (*q,
δD*)$_{SCL}$, which subtracts out the SCL variations (Fig. 13). CL data are taken primarily from LL and SAW modules
(processed as in Section 2.4). For these modules, (*q, δD*)$_{SCL}$ were taken from temporally neighboring data below 500
m, which usually occurred within 10 minutes before or after the module, but in some cases ± 20-minute windows
were necessary. CL data were taken secondarily from vertical profiles (Section 2.1) which did not include LL or
SAW modules. For these profiles, (*Δq, ΔδD*) were computed from the vertically-averaged values of the SCL and CL
for the respective profile. Only CL data for which the magnitude of the mean relative wind between the SCL and CL
was less than 3.5 m s$^{-1}$ were used to avoid noise due to large relative horizontal advection. Each block of SCL data



used for computing $(q, \delta D)_{SCL}$ was also used to compute mean cloud-condensation-nuclei (CCN) values. Due to
instrument limitations it is necessary to use SCL rather than CL CCN measurements, but Diamond et al (2018)
showed that for the 2016 IOP, SCL CCN correlates well with CL droplet number concentration, implying
connection between the SCL and CL aerosol concentrations.

Figure 13 shows $\Delta q$-$\Delta \delta D$ diagrams colored by SCL CCN tercile groups. The CCN terciles are different for each year
and are listed in Table 3.3. Distributions of SCL CCN are shown in Fig. 14. The solid and dashed black curves are
mixing and Rayleigh curves. The endmembers for the mixing curves are $(q, \delta D)_{SCL}$ and the same dry, isotopically
depleted endmember used in figures 7-9. From here on out this mixing trajectory is referred to as $mix_{FT}$. Data which
fall below $mix_{FT}$ are candidates for precipitation but there are two possibilities. The first is that the CL has a recent
history of precipitation. The other is that the air entraining in at CL top tends to align the data along an alternate
mixing trajectory. To test this, mixing curves between $(q, \delta D)_{MBL}$ and the mean $(q, \delta D)$ of the 250 – 500 m region
above CL top $(q, \delta D)_{250-500}$ for all vertical profiles in Sect. 2.3 are included in Fig. 13 and from here on out are
referred to as $mix_{250-500}$. They are expressed as $(\Delta q, \Delta \delta D)$ like the scatter points. The dark portion of the curves
represent mixtures that have at most 25 % $(q, \delta D)_{250-500}$ contribution. CLs sampled during the 2016 IOP occurred at
higher latitudes in comparison to the other two IOPs and the CLs were on average 2-3x shallower (Fig. 14). The
majority of CLs show small deviations ($\Delta q < 0.5$ g kg$^{-1}$, $\Delta \delta D < 4$ ‰) from $(q, \delta D)_{SCL}$. The majority of $mix_{250-500}$
curves (30 out of 35) are centered about $mix_{FT}$ and clearly depart from the Rayleigh trajectory. Five of the curves fall
closer to the Rayliegh trajectory.

In comparison to the 2016 IOP, a higher proportion of 2017 IOP data deviate from $(q, \delta D)_{SCL}$, and the standard
deviation is larger. There are fewer data points shown for 2017 - since three of the flights were eastward toward
Ascension Island, days where the MBL has transitioned to shallow cumulus rather than a SCMBL were omitted.
Two thirds of the data lie to within precision of one of the mixing curves or the Rayleigh curve. The other third
cannot be described by the simple models presented here. As in the 2016 IOP, the majority of $mix_{250-500}$ curves are
clustered about $mix_{FT}$ (19 of 24) while 5 pass near the Rayleigh trajectory.

For the 2018 IOP, the alternate endmember mixing curves cover a wide range of $(q, \delta D)$ space rather than being
clustered about $mix_{FT,}$ which likely explains the middle tercile CL data (yellow). Like in the 2017 case, the data
which lie near a Rayleigh trajectory with $\Delta \delta D > 5$ ‰ (blue cluster) could not be explained by mixing curves unless
the CL contained a fraction of FT air greater than 25 % in comparison to the SCL.



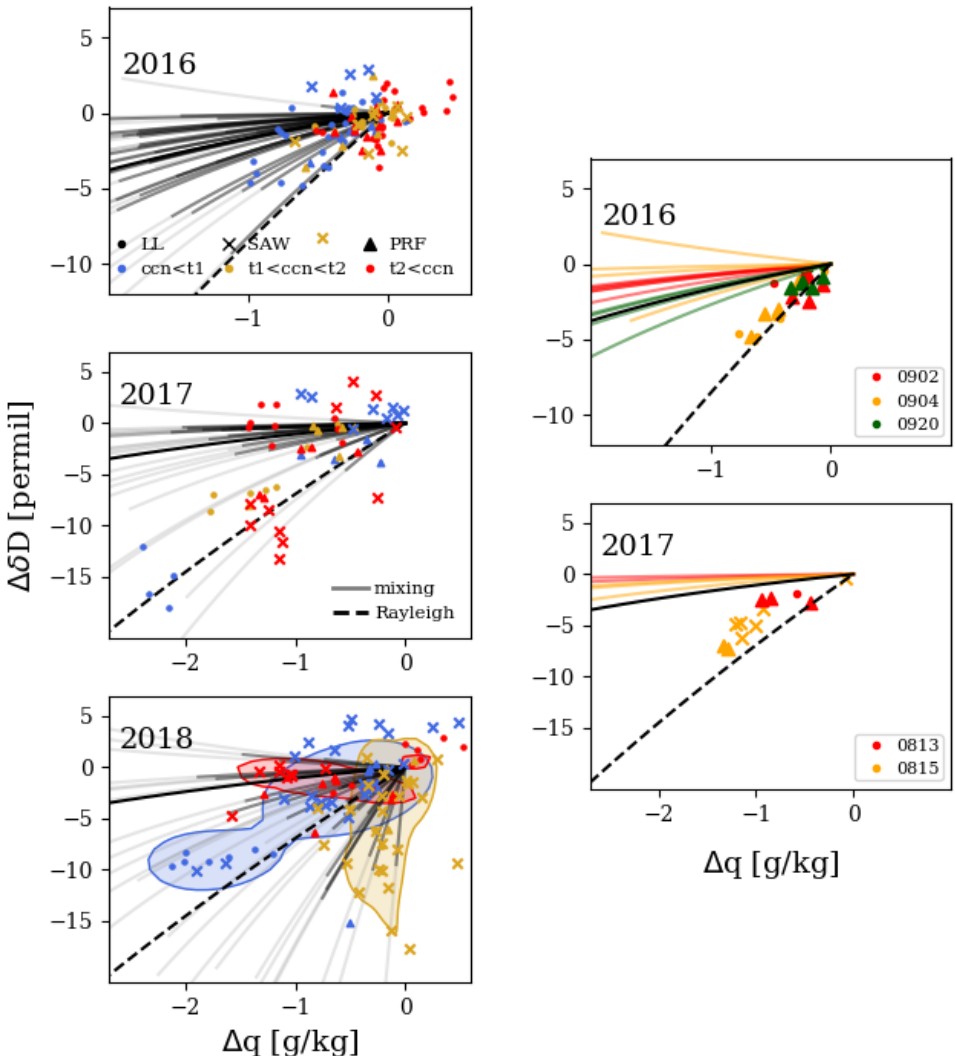

**Figure 13. (Left column) CL (*q*, *δD*) data expressed as deviations from the nearby SCL mean values, as described in the main text. Symbols denote the flight module types level leg (LL), sawtooth pattern (SAW) and vertical profiles (PRF). Colors denote tercile groups for SCL CCN (terciles listed in Table 3.3). (Right column) Data from several flights during the 2016 and 2017 IOPs are isolated. These data fall near a Rayleigh trajectory and cannot be explained by mixing with the overlying FT measurements for that day.**





**Table 3. Sub-cloud CCN (cm⁻³) terciles for the groupings in Fig. 13.**

| IOP year | min | tercile 1 | tercile 2 | max |
|----------|-----|-----------|-----------|-----|
| 2016 | 19 | 134 | 221 | 426 |
| 2017 | 41 | 186 | 357 | 926 |
| 2018 | 16 | 188 | 250 | 460 |


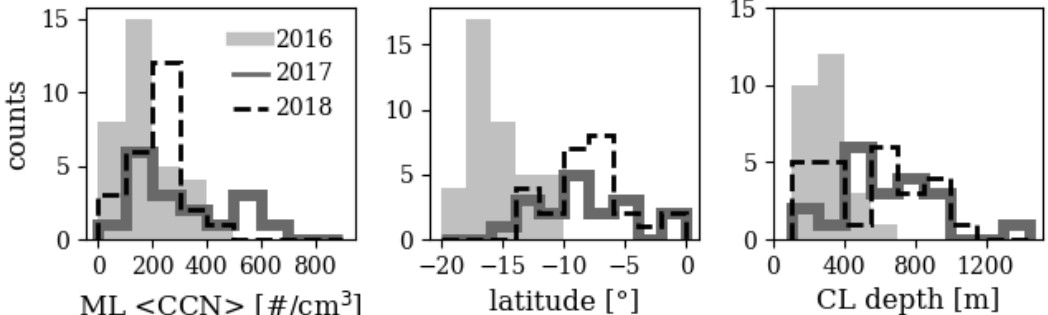

**Figure 14**. **Histograms of SCL CCN (left), CL sampling latitudes (middle), and CL depths (right) for each IOP.**

## 5 Discussion

*Sub-cloud layer δD*

In this study the sub-cloud layer $\delta D$ has been assessed, briefly, for agreement with basic theory as well as typical variations. Agreement with theory has been assessed on two different scales: the coarser scale of Figure 8-10 and the finer scale of Figure 5. For the former, it was confirmed that to first order, median values for each IOP lie within 7 ‰ to mixing curves between vapor at equilibrium with a mean SST and dry FT air, giving the expected result that

the primary moisture source for the SCL is ocean evaporation as opposed to evapotranspiration from a land source. However, to advance theory of oceanic isotopic exchange observational constraints better than 7 ‰ are needed (e.g. Benetti et al., 2018; Feng et al., 2019). A more detailed comparison of the data with an MJ79 prediction shows disagreements of 10 ‰ for individual data points. A large source of variation for the predictions is the assumed $\delta D$ of the ocean for which there are no measurements during ORACLES. Using Fig. 4 of Benetti et al. (2017) as a

guide, an ocean surface $\delta D_{oc}$ of 0 - 6 ‰ is reasonable for the 2017 and 2018 IOPs. The MJ79 predictions using 6 ‰ provide a rough upper bound on the SCL measurements for 2017 and 2018 (Fig. 5), which makes sense since FT entrainment and precipitation could decrease SCL $\delta D$. For 2016, even a $\delta D_{oc}$ of 10 ‰ does not provide an adequate



upper bound. One possibility is that $\delta D_{oc}$ is even higher than estimated. Another is the WISPER system calibration uncertainty to the absolute scale, which is comparable to the difference from this campaign (see Henze et al., 2022).

Additionally, variability throughout the SCL column has been assessed, which is novel since data above the surface (e.g. sampling from ship masts ~30 m) are uncommon. The SCL is characterized in part by nearly constant vertical profiles of quantities such as water concentration and potential temperature, and similarly the isotope ratios are expected to as well. The right column of Fig. 5 shows that this is often the case for $\delta D$. Vertical deviations over the layers are small, typically less than 2 ‰. These deviations are smallest for 2016, which is reasonable given the

shallower SCMBLs would likely be more coupled to the surface. Additionally, the speed of the aircraft affords a snapshot of an SCL's horizontal variability, as opposed to surface (ship or island) based measurements which must rely on advection. Horizontal P-3 legs show that even over distances of 50 km (6 minutes of P-3 travel time), 10 s averages of $\delta D$ typically do not vary by more than 2 ‰.

*FT and BBA plume hydrologic histories*

Figures 6 and 7 show that FT air from the 2016 sampling period experienced the least amount of precipitation while over Africa of the three sampling periods. The in-situ ($q, \delta D$) data show that in fact there is almost no signs of precipitation in the sampled high-BBA air in 2016. The center of the high-BBA PDF falls almost directly on a mixing curve between the center of the SCL observations and dry FT air, and the orientation of the PDF follows the mixing trajectory well. The PDF is enveloped from above by the theoretical ET mixing curve, suggesting ET

contributions for some of the measurements, although the dominant signal is simple mixing with air close to the observed SCL values. The low-BBA data are shown for completeness, but since back-trajectories suggest they often originate from the southern Atlantic rather than Africa, they are not analyzed in detail.

Both high-BBA and low-BBA data for the 2017 IOP show signs of precipitation (Fig. 9). The data are bound from above by a mixing curve with a moist end member including 25% ET contribution and from the right by a family of

Rayleigh curves. The data support a depiction of the air mass hydrologic history as convective detrainment followed by dilution; airmasses follow the Rayleigh curve during moist convection (i.e., condensation with high precipitation efficiency), then begin to detrain remaining total water at their level of neutral buoyancy, and subsequently form a diluted mixture with surrounding dry FT air. The low-BBA air is associated with convection which reached higher altitudes. A correlation of -0.67 for $\delta D$ vs. height is obtained, whereas the correlation is close 0 for the 2016 IOP

despite sampling at a similar altitude range. For the 2017 IOP, detrainment heights and temperatures predicted from the Rayleigh model both have correlations of 0.67 with the observations, which is remarkable considering the simplicity of the parcel-type model used to describe the entire study period: That is, a single choice of values for surface $T$, $q$, and height, no entrainment during convection, and all condensate converted to precipitation provides a reasonable approximation. The best fit lines have slopes of 0.54, however. Possible reasons why this slope is less

than 1 include variation in surface conditions during convection throughout the observation period, and precipitation efficiency less than unity. With respect to that latter, entrainment into an updraft would decrease $\delta D$ more than expected from the Rayleigh process alone. Therefore, to achieve the same $\delta D$ with only a Rayleigh process, the



parcel would have to be lifted higher to compensate. Additionally, $\delta D$ along detrainment curves was taken as constant to simplify the calculations. However, $\delta D$ decreases by about 20 ‰ once detrained updraft air has been
diluted to around 2 g kg$^{-1}$, resulting in the same bias for the Rayleigh model as precipitation efficiency < 1 does.

The Oct. 2018 sampling period comprises elements of both the 2016 and 2017 sampling periods. The high-BBA comes from higher latitudes similar to 2016, while the low-BBA air is further north with more rainfall. However, unlike 2016, the high BBA air flows northwest over Africa before flowing offshore. The high-BBA air $(q, \delta D)$ signal reflects this, the PDF lobe falls along a mixing line with evidence of up to 25 % ET contribution, but the
distribution extends below the mixing line much more so than in 2016, bounded beneath by detrainment at 3600 m. Meanwhile, a large fraction of the low-BBA data lies near or below a Rayleigh curve, suggesting convection with a high precipitation efficiency and little subsequent mixing and potentially re-evaporation of falling droplets. Data with this signal are almost exclusively at heights below 3000 m, while data above 3000 m either fall directly on a Rayleigh curve or in the region hypothesized to be detrainment. Perhaps rain droplets falling from these higher
altitudes are partially evaporating into the air below 3000 m. Such refined hydrologic history is not readily extracted without more detailed modeling.

*Parsing entrainment mixing vs. precipitation in the CL and detection timescales*

Cloud layer data for the 2017 and 2018 IOPs deviate further from their respective $(q, \delta D)_{SCL}$ values in comparison to the 2016 IOP. This is in line with the deeper CLs (Fig. 14) transitioning to cumulus coupled CLs, becoming less
well-mixed with the SCL, and experiencing higher entrainment and precipitation rates. CL $(q, \delta D)$ measurements for all IOPs are poorly constrained by the simple $mix_{FT}$ and Rayleigh models – around 50 % of data for the 2016 and 2017 IOPs lie within these models, and only 30 % for the 2018 IOP. For the 2018 IOP, the majority of data outside these bounds can be explained by mixing with air masses above the MBL which have varying $(q, \delta D)$ values. For the 2016 and 2017 IOPs, even alternate endmember mixing cannot explain the data and these exceptions merit
further study since the fraction of the data they make up is non-negligible. One possibility is the choice of a single representative $(q, \delta D)$ pair for the SCL when generating the $mix_{250-500}$ curves. If either or both of $q$ and $\delta D$ for overlying air were higher than the SCL values, the resulting mixing curves could explain the data in question. Differential horizontal advection of the MBL and FT could also be a contributing factor.

Although the data show signals of local precipitation in the 2016 and 2017 IOPs, the null hypothesis that all data
falling below the $mix_{FT}$ curve came from $mix_{250-500}$ curves cannot be fully ruled out with this analysis alone. However, Fig. 13 (right column) highlights some cases where the data fall near a Rayleigh trajectory and cannot be accounted for by mixing models with overlying FT air sampled on those days. Previous work has highlighted the uncertainties involved with assuming an MBL is in significant contact with the FT air above it at the time of sampling and neglecting horizontal advection (Diamond et al., 2018). Therefore, it cannot be definitively concluded
that the cloud layers sampled during the highlighted flights recently precipitated. However, these flights would be good candidates to examine with advanced modeling.



Assuming that at least some of the signal is due to precipitation, the question arises regarding the time period prior to the measurement during which the precipitation occurred. Simple bulk model studies (Schubert et al., 1979; Bretherton et al., 2010; Jones et al., 2014) estimate a thermodynamic re-equilibration time scale of a SCMBL at 0.5 -
1 day. Therefore, at longer timescales it is assumed that any isotopic evidence due to precipitation has been 'erased' by subsequent turbulent mixing, including influences from surface evaporation, and consequently any signals detected are from precipitation that occurred within the past day. Previous observational studies show SCMBLs precipitate primarily at night or the early morning hours (Burleyson et al., 2013, Leon et al. 2008, Comstock et al. 2004). This argument is consistent either the weaker signals observed in the SCMBLD in the ORACLES dataset. It
may be the case that the majority of precipitation signals detected with isotope measurements are from the morning on the same day as sampling.

*(q, δD) signals and SCL CCN*

By inspection, there is weak SCL CCN clustering in Fig. 13 for the 2016 IOP. The two higher CCN tercile groups cluster about the origin while the lowest CCN tercile group extends to more negative values. If only a fraction of the
signals come from $mix_{25-500}$ processes, the data would indicate that aerosol loading has induced precipitation suppression. The converse could also be the case: aerosol scavenging in SCMBLs with previous precipitation result in lower SCL CCN. Microphysical modeling would be required to quantify the causality. Further, no clear clustering is present for the 2017 IOP, in part due to the smaller signals. Indeed, using either thermodynamic or isotopic measurements to assess cloud aerosol interactions in SCMBL is not necessarily achievable with simple analytical
models. As a case in point, one confounding factor could be locally enhanced entrainment due to precipitation (see e.g. Fig. 11 in Wood, 2012), which could replenish BBA aerosols which were scavenged.

For the 2018 IOP, a consistent explanation develops from Fig. 10 and Fig. 13 (which present the 2018 IOP FT and CL data respectively). CL data in the middle and top SCL CCN tercile groups are largely clustered around $mix_{FT}$ or the region below the Rayleigh curve (Fig. 13, shaded regions). Of these data, the highest tercile group is primarily
along the former while the middle group is along the latter. This is consistent with the (*q, δD*) location of high vs. low BBA FT PDFs in Fig. 10, if they were to serve as the above MBL mixing endmembers. Therefore, the CL data in the middle and top tercile groups are likely both mixtures of SCL and FT air, with the differences in MBL CCN due to the differences in FT BBA levels. Data in the lowest tercile cluster closer to a Rayleigh trajectory, possibly indicating local precipitation and aerosol scavenging. This self-consistency in the joint *q-δD* measurements builds a
degree of confidence that CL isotopic signals can be connected to entrainment processes, and potentially precipitation, even if the connection is more detailed as for the 2017 IOP.



## 6 Final Remarks

The ORACLES FT mass histories are deduced using the process-based expectation of variations in $q, \delta D$ space, described by simple analytical models, in combination of several other data sources (satellite, reanalysis, and GCM output). The graphical framework is convenient to develop quantitative estimates of moisture contributions (e.g. fraction of moisture from ET and convective detrainment heights). The analysis partitions airmasses into those which enter the FT via dry convection/vertical-mixing vs. via convection accompanied by precipitation. In 2016 the BBA air entered into the FT almost entirely by vertical mixing: likely exported latterly from the continental PBL, with dilution of FT air aloft. In contrast, in 2017 both high and low BBA FT air are primarily associated with convective detrainment, where measurements provide evidence that low-BBA air is associated with convective detrainment at higher altitudes. In 2018, the lower BBA air experienced precipitation. In fact, most measurements indicate that direct convection (e.g., minimal dilution after detrainment) was sampled, and considerable fractions of low-BBA air lying below the Rayleigh curve is indicative of partial re-evaporation of falling rain drops. High-BBA air for 2018 shows a combination of mixing with ET contribution and convective detrainment which is important to the hydrologic picture.

There is strong correlation between ($q, \delta D$) evidence of precipitation and BC/CO evidence of scavenging. As a control, BC/CO along mixing trajectories show little consistency. Change in $q$ due to precipitation, $\Delta q_{precip}$, derived from the isotopic data is correlated with BC/CO, 0.71 for 2017 and 0.56 for 2018. This diagnostic partitioning cannot be achieved with the humidity measurements alone - correlation between $q$ and BC/CO is 0.21 for 2017 and -0.14 for 2018. Some previous studies on black carbon scavenging use BC properties and BC/CO ratios in cloud droplets (Ohata et al., 2016; Liu et al., 2020; Twohy et al., 2021) to assess scavenging mechanisms and coefficients. However, they are unable to link their results to the integrated precipitation amount. Here, because the isotopic data serve as an independent measure of integrated precipitation (rather than a precipitation rate as with radar), they may provide a unique and complimentary way of determining BC scavenging in the FT. For the analysis presented here, BC concentrations in the African PBL would be required to derive numerical values for the scavenging coefficient. An ideal sampling strategy would observe the $q, \delta D$, and BC of air within and above the PBL simultaneously measured in a Lagrangian framework, providing direct values of $\Delta q_{precip}$, the change in BC, and in turn a scavenging coefficient.

An analysis utilizing relative differences between the SCL and CL was chosen due to large measurement uncertainties and lack of ocean surface $\delta D$ measurements. This posed challenges since relative differences in $\delta D$ between these two vertical regions tend to be small. While no strong statements on precipitation or their interaction with aerosols can be made from this study, several key conclusions are made. For all IOPs, the fact that the CL is coupled to the surface will tend to erase signals of precipitation events, or at least convolute them with other process. From previous studies, it is inferred that the time scale at which this occurs to be 0.5 - 1 day. However, by far the most difficult obstacle for the ORACLES dataset is the ambiguity in CL signals due to variability in ($q, \delta D$) of the entraining air. Some air masses sampled above cloud top had values that, if entrained into the CL, were capable of



lowering the CL $\delta D$ values towards a region of ($q$, $\delta D$) space that would otherwise appear as precipitation. Such airmasses were observed about 14%, 20 %, and 60 % of the time for the 2016, 2017, and 2018 IOPs respectively. For the 2016 IOP, the data which do suggest recent precipitation are typically connected to MBLs with the lowest SCL CCN values, which could reflect aerosol scavenging. For 2018 IOP, the CL ($q$, $\delta D$) data is combined with SCL CCN values and the FT ($q$, $\delta D$) diagram to summarize the regional conditions: the middle and highest CCN values

in the MBL are likely connected with variation in aerosol levels of entraining air, while the lowest MBL CCN values are the result of local precipitation. It may be the case that simply looking at relative differences between the SCL and CL does not produce an adequate signal, but that an MBL which experiences precipitation, even with subsequent mixing, can produce a column-averaged $\delta D$ lower than that for an MBL with only mixing and no precipitation. More complex models appear necessary to obtain constraints on precipitation versus entrainment

mixing and their effects on MBL aerosol concentration.

**Appendix A: Estimating evapotranspiration contribution to total water mass of a boundary layer column**

Because ET contribution can alter the isotopic content of an airmass, its expected fraction of moisture contribution for each sampling period was estimated using the following back of the envelop calculation. For a marine boundary

layer column with cross-sectional area 1m$^2$, the total mass of water in the column $m_{H2O,col} \sim hq_a\rho_a$, where $h$ is the height of the column, and $q_a$, $\rho_a$ are the column-average specific humidity and air density. If the column advects onto land, then the total mass of water added to the column by ET over some time $\Delta t$ is $m_{H2O,ET} = (ET)\Delta t$, where $ET$ is the mean net evapotranspiration in units of $\mathrm{kg\,m^{-2}\,s^{-1}}$. Therefore, assuming negligible input of water from free tropospheric entrainment, the fraction of ET contribution is $m_{H2O,ET}/(m_{H2O,ET} + m_{H2O,col})$. ET is estimated using the

MODIS/TERRA Version 6 Evapotranspiration/Latent Heat Flux product (MOD16A2), taking the mean and standard deviation over the boxed regions of Fig. 5.

**Author Contributions**

DH carried out the formal analysis and visualization, and prepared the manuscript with contributions from all co-

authors. DN obtained funding for the project, contributed to analysis conceptualization, and provided supervision. DT provided resources and critical guidance to place the analysis in the context of aerosol and black carbon research.

**Competing Interests**

The authors declare that they have no conflict of interest.



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
