# Peer review of "Detection of dilution due to turbulent mixing vs. precipitation scavenging effects on biomass burning aerosol concentrations using stable water isotope ratios during ORACLES"

_EGUsphere, 2023_

## Referee Comment (RC1)

**Review of "Detection of mixing and precipitation scavenging effects on biomass burning aerosols using total water heavy isotope ratios during ORACLES" by Henze et al.**

This paper presents very innovative research on the use of water isotopes to assess the impact of different atmospheric processes on biomass burning aerosols using observations from the NASA ORACLES field campaign. I enjoyed reading this very interesting manuscript! I have no major scientific comment, the work is of high quality, but I would strongly recommend the authors to refine the writing and presentation of their results (figures quality). Given the wide range of tools and expertise that is combined in this publication, more clarity in the writing is essential to make these valuable findings more accessible for both isotope and aerosol experts. My minor comments, which you'll find below, as well as a small number of technical points listed at the end of my review will hopefully help the authors to refine the presentation of their work.

Major comments: improve the readability of the text and the reader guidance as well as the quality of the figures.

Minor comments:
1) Title: with "mixing" it is not entirely clear what you mean, would it make sense to point out here already the dry vs. moist convective mixing pathway of biomass burning aerosols (BBA) into the free troposphere (FT) that you are addressing in the paper?
2) L 11-12: at this stage it is not clear to me that you mean dilution by turbulent mixing in the free troposphere. Maybe it would help me if you wrote "precipitation scavenging" and "dilution due to mixing". And in the next sentence, I would keep the same order of processes as in the first sentence: precipitation processes versus mixing.
3) L. 15: "Air… is distinct, and can be treated as separate analyses…" After reading the full paper I now understand what you mean but without knowing the paper this sentence sounds a bit mysterious. In which respect are FT and MBL air masses different?
4) L. 15: since you don't use MBL again in the abstract you don't need this abbreviation there, only in the main text.
5) L. 16: remove "to"
6) L. 17: "assess of…" I don't understand this sentence, something went wrong with the phrasing here. Maybe you mean "are used to assess the relevance of the air parcels' precipitation history"?
7) L. 19: make clear that MERRA-2 is a global reanalysis dataset
8) L. 28: CCN is not defined in the abstract
9) L. 31: Again make clear that you mean dilution by mixing: "signature of dilution effects by air mass mixing in the free troposphere". In my first reading I thought you mean dilution of heavy isotope concentrations by rain out underway. I didn't immediately think of mixing. But maybe making this clear at line 12 is enough. Of course, when reading your paper, one wonders what process exactly leads to dilution in the free troposphere. The turbulence is likely very limited except in the environment of clouds or in regions with strong vertical or horizontal wind shear…

10) L. 41: "particularly after again"? There seems to be part of the sentence missing here.

11) L. 44: "…quantifying processes… is of importance for climate models."

12) L. 54: maybe "compared to the reduction…"

13) L. 54: this is the first time the term dilution comes up in the main text, make clear that you mean dilution by mixing through turbulence and convection.

14) L. 57: how about first saying: "they provide information on the relative importance of the history of moist atmospheric processes such as mixing…"

15) L. 60: "While isotope ratio information to date has shown promise…". This is true but a bit unspecific.

16) L. 62: "… for the advantages to be realized". Maybe "to take full advantage of the tracer capacity of water isotopes and their use for assessing aerosol cycling in the atmosphere".

17) L. 67: "The isotopic compositions for water D/H" is a pleonasm. Maybe "the isotope composition of water" or the "concentration of HDO"?

18) L. 70-71: "In the case of the FT, regional aerosol and moisture transport are connected and characterized" what does "connected" mean exactly? That they are considered together?

19) L. 82: How are the relative importance of evapotranspiration, dry versus moist convection etc. constrained? This needs to be explicitly clarified and the (q,dD) phase space is so essential for your paper that it needs to be introduced in the introduction.

20) L. 69-90: this whole section is a bit unspectacularly written and sounds more like a casual report or list of things that were done in this paper (e.g. "Next,… is evaluated."), but in a way that is not rising the readers curiosity and attention. This paragraph is absolutely key and it is very cool! I think the authors could invest in building up some tension here, maybe listing the research questions, rising the reader's curiosity.

21) L. 88: A new paragraph could be started at "Section 2 covers…".

22) L. 89: "Section 3 presents a brief **analysis** of the sub-cloud as a starting point for contrasting the **analysis** of the FT **analysis** of precipitation histories and scavenging". Please rephrase.

23) L. 91: "precipitation metrics", not sure what you mean here, the ones derived in Section 3?

24) L. 106: "Flights were typically every 2-3 days, which 7-9 hours in duration and…".

25) L. 114: I would leave away "Measurements of" and just write "Potential temperature…".

26) L. 146: "Using vertical profiles…" I don't fully understand this sentence. In particular, why there is a logical link between "using RH over liquid water as a flag for the CL" and "profiles through CLs with broken clouds".

27) Section 2.5 first introduce the (q, dD) diagram, it is otherwise a bit difficult to follow and understand what is meant here.

28) L. 171: What do you mean by "subsequent mixtures between them"?

29) L. 187: There are no theoretical lines in Fig. 3, you probably mean Fig. 4.

30) Fig. 3: is there a colormap missing or is it just that you color the points in the total water mixing ratio?

31) L. 201: Rephrase, "for air where moisture from the respective evaporation source…" there is a verb missing or "and" is not needed.

32) L. 203: you probably mean the blue circle in Fig. 4.

33) L. 208-220: In this paragraph one can get lost as an isotope beginner. Guide the reader, what is the aim? Also, it is not entirely clear which variable is measured which is modelled/taken from the reanalysis and what the uncertainty resulting from the use of the MERRA data is.

34) Section 3: the start of the results section is a bit abrupt with a detailed description of Figure 5. The reader could be led a bit more smoothly through the results and its structure. The different results sections come a bit as a surprise.

35) L. 227: Rephrase the first sentence, something went wrong. Maybe remove "of"?

36) L. 230: nice that you refer to previous studies, but this could be done in the intro or the discussion, start with addressing your question. I think Feng et al. 2019 didn't present new observations.

37) Results: From a structural point of view, and given your research questions sketched in the introduction, I am not convinced that starting with the subcloud layer is a good idea. But maybe there is a good reason for this, then state it to make sure the reader knows why he/she is reading this.

38) Results and discussion: Given the complexity of this study, I very much doubt the use of separating the results and discussion section. Now that I made the effort of understanding your very interesting Fig. 5, I would like to learn what it tells me about moist processes in the boundary layer. Until Section 5 (where you discuss the science), I will have forgotten most of what you show in Fig. 5 because I will have had 10 additional complex figures to understand.

39) Fig. 5: When I look at Fig. 5, I don't really see these measurements as fitting particularly well in a mixing framework or along an imaginary mixing line.

40) L252: what is the output frequency of the interpolated air parcel positions? Are hourly 3D wind fields used? The information on the trajectory calculation setup belongs into the methods section.

41) L258: "double that of the SCL" I don't' understand this. What is the CO concentration of the SCL, is it shown somewhere?

42) L262: remove it, "The low-BBA air in 2016 is…"

43) L285ff: Make clear that this assessment with (monthly?) surface precipitation collocated with trajectory information is a first coarse assessment of the possibility of precipitation formation underway. It is not clear if 1) precipitation did really form at the time of the airmass overpass over a given region (timing uncertainty) and 2) if the airmass that you are tracking is really the one involved in precipitation formation, it could be that it is travelling above the cloud (vertical level of precipitation formation uncertainty)

44) L300ff: The sources of water vapour and BBA are likely not the same… This should be critically discussed somewhere. Did the authors look at forest fire observations from remote sensing to identify potential BBA sources?

45) L347ff: I am not sure, I understand why correlations are computed between dD and altitude, and in which context to place the vertical gradients of dD. Here the reader needs help to understand why these analyses are done.

46) Table 3: Was isoCAM evaluated with respect to near-surface observations over tropical Africa? How confident are the authors that this data is reliable? How is the

treatment of ET done in isoCAM? Is there a multilayer soil model to take near-surface soil layer enrichment due to evaporation? Is soil evaporation and plant transpiration treated in a fractionating or non-fractionating way?

47) L385: why was this data discarded? It would provide a very interesting control group and reference point in the dD-q phase space for "clean" airmasses with likely important moisture contributions from the MBL.

48) Figs. 9 and 10 are complex and thus a bit too small to understand well.

49) In Fig. 11 many data points seem to be overlying each other. This makes the figure difficult to read.

50) Fig. 11 and 12 and related results: I really like this approach, but I got lost when trying to understand how the convective plume model was exactly set up for these calculations. There are several unconstrained parameters in this model. I think a dedicated methods section on the setup of the convective plume model would really help me to see through the approach.

51) To me turning back to the SCL and Fig. 5 now feels like quite a slalom.

52) L. 449: CCNs are already introduced as an abbreviation, define it above when first mentioning them in the main text.

53) L496: As already mentioned earlier I find this part about the subcloud layer a bit weak. The motivation to assess the dD "for agreement with basic theory" is not very convincing. I like the idea of assessing the horizontal and vertical variability of the dD in the MBL and it would be nice to have 1 or 2 short statements about where this variability comes from process-wise.

54) L520ff: I am confused by the reasoning behind the first paragraph of this subsection: why should the high-BBA PDF be linked to the SCL observations? I thought that the hypothesis is that the transport pathway of moisture into the free troposphere is: 1) ET over tropical Africa, 2) dry or moist convective detrainment into the free troposphere, 3) Advection over the South Atlantic. So, if there is a link with the SCL measurements in the MBL that would be by convective downdrafts bringing dry FT air into the SCL, right?

55) L558: I can't really see that in Fig. 13, to me it seems that many data points in all three IOPs deviate a lot from the mixing and Rayleigh lines.

56) L560ff: I must have missed it but is the MBL defined as the SCL and the CL in your study? Could that be included in the sketch in Fig. 1.

57) L568: that's very interesting!

58) L611: Repeating the fact that three IOPs over three years in the late winter period August-October with xx number of flights and in total yy hours of measurements were involved would be great.

59) L612: in combination with?

60) L613: "of moisture contributions from different processes along the atmospheric transport pathways of water vapour (e.g….)"

61) L615: "accompanied by cloud formation and precipitation…"

62) L625 & L626ff: "This diagnostic partitioning cannot be achieved with humidity measurements alone…" very nice result indeed!!

63) L655: It would be fantastic if the paper could end with a more general closure statement of why this combination of isotopes with aerosol data is useful and why research on this combination of observations should be further pursued in the future.

Technical comments:
- The Figures are all very interesting, but the quality (resolution?) is a bit poor. It's a pity, if it stays like this because of their very valuable content.
- There are many instances with excessive spaces, double .. or ,, or missing commas.
- To make this complex paper (in terms of the number of aspects and processes involved) easier to access to the readership, I would strongly recommend reducing the number of abbreviations. There are some that are clearly needed, but others are used only a few times and could be written out. For example, for the boundary layer there is SCMBL, MBL, PBL… do you really need all of them or could you just abbreviate BL and then write out the specific one you mean? Or for example biomass burning (BB) you don't use so many times, you mainly use BBA, so could you write out "biomass burning" in the few BB instances? I think similar considerations for other abbreviations would be worthwhile to make the paper easier to read.
- Many abbreviations are not properly introduced in the text, they should be written out the first time they appear (PBL, MBL, LCLT, CCN in the abstract).
- Find a consistent term to refer to the (q,dD) phase space, it is sometimes (q,dD) and at other times q, dD. Also, this is such an important tool for the paper, could it be mentioned already explicitly in the abstract?
- All variable names in italics, e.g. $R_D$ (L. 113). RH is sometimes in italics sometimes not (e.g. L. 152 and L. 155). However D as a chemical notation for deuterium should not be in italics.
- Airmass is sometimes written in one word sometimes in two (air mass). The same for end member vs. endmember.
- References mentioned directly in the text like Fiorella et al. (2022) have no comma: E.g. not Fiorella et al., (2022) at L. 316.

---

## Referee Comment (RC2)

The mixing and precipitation scavenging effects are difficult to be quantified in observations. The authors showed a promising way by analyzing the observations of water isotope ratios and humidity together. The authors successfully show the difference between the mixing and scavenging processes with the new campaign data in the southeast Atlantic and their humidity-isotope pairs analyses. I believe the method is novel to the field of the interaction between clouds and aerosols.

I don't have major scientific comments for the manuscript, though I think the manuscript should be proofread again. Since the topic of this manuscript is the biomass burning aerosols, I think the manuscript should highlight them more in the text. There are a lot of texts in the manuscript describing the convection processes. The authors can link them to the concentration of BBA.

Minor comments:

Line 16: remove "to"?
Line 17: remove "of "
Line 28: What does IOP mean?
Line 41: after again?
Lines 52-53: "A component of both ….". The sentence reads weird. Why is the component of BC lifetimes the aerosol-cloud interaction?
Line 70: Double "."
Line 71: Double ","
Line 78: double "topped"
Figure 1: It would be nice to put two subplots in one row.
Line 96: Missing period after Redemann et al., (2021).
Line 98: What does LCLT mean?
Line 101: deck due to -> deck is due to
Line 101: "where it": what does where refer to? Subsidence or cloud deck? What does "it" refer to?
Line 107: missing spaces in "100m" and "7km"
Figure 2: Please add (a)(b)(c) for each subplot. Please choose a different color scheme for the left panel. It's not easy to tell three different tracks.
Line 146: "Using vertical profiles of T… was chosen": Please check the grammar here.
Line 147: "it would also…": What does it refer to?
Line 175: qdD is very important for this paper. Could you elaborate why qdD is useful than theta_e and what does it mean? Is it q times dD or something else?
Figure 3: What do two profiles in the middle of Figure 3(a) represent? What does the multi-color mean? Should it be accompanied with a color bar?
Figure 209: Where are the "grey" curves? In Figure 3?
Figure 228: What is the spatial resolution of this COBE dataset? What does COBE mean? References?
Line 257: "low-BBA are": double "are" in this sentence.
Figure 6: What is the difference between the dashed and solid line boxed regions? Also, no colors are filled in the color bar.

Line 303: What's the resolution of this MODIS data? Please briefly introduce this dataset, including its duration.

Line 304: Which boxed region?

Line 311: Table 3.2 or Table 2?

Line 317: 2022-> 2021

Line 323: figures -> Figures

Table 2: The row of "ocean evaporation source", the column of "calculation": equations 6-> Equation 6. "Horita and Wesolowski, 1994": Is it the correct citation style?

A weird "345" appears in the last row and last column.

Line 361: blue shaded region: light blue?

Line 363: Table 3.2 -> Table 2

Line 384: What does rBC mean?

Line 415: Two "is" in one sentence.

Line 431: Please reduce the dot size to increase the visibility, or change this figure to a heatmap.

Line 456: figures -> Figures

Line 501: add "the" between "advance" and "theory".

Line 534: close 0-> close to 0?

Line 541: efficiency less than unity -> efficiency is less than unity

Line 575: "However": Two however in one paragraph.

Line 584: I don't get what "either" means here.

---

## Author Comment (AC1)

Authors' response to Referee #1's review of "Detection of mixing and precipitation scavenging effects on biomass burning aerosols using total water heavy isotope ratios during ORACLES" by Henze et al.

**Dear Referee,**

**Thank you for your thorough and thoughtful review, as well as your encouraging remarks! Find our responses in bold below each of your comments/suggestions**

This paper presents very innovative research on the use of water isotopes to assess the impact of different atmospheric processes on biomass burning aerosols using observations from the NASA ORACLES field campaign. I enjoyed reading this very interesting manuscript! I have no major scientific comment, the work is of high quality, but I would strongly recommend the authors to refine the writing and presentation of their results (figures quality). Given the wide range of tools and expertise that is combined in this publication, more clarity in the writing is essential to make these valuable findings more accessible for both isotope and aerosol experts. My minor comments, which you'll find below, as well as a small number of technical points listed at the end of my review will hopefully help the authors to refine the presentation of their work.

Major comments: improve the readability of the text and the reader guidance as well as the quality of the figures.

**In regards to your two high level concerns:**

**Readability:**
**The abstract has been restructured and reworded to address some of the minor comments but also to provide more clarification of the study's focus and results. Vague statements are replaced by more explicit ones. Sections 2.6 and 3.1 have been moved to appendices, keeping the paper's focus on the main analysis. Not having to address section 3.1 in the discussion also streamlines that section.**

**Figure quality:**
**The figures look high quality and crisp in the web viewer version of the PDF for the preprint here: https://egusphere.copernicus.org/preprints/2023/egusphere-2023-69/egusphere-2023-69.pdf . But once downloaded some of the figures are blurred. We will keep an eye on this, but are expecting it to resolve itself for the final submission, as we will be submitting the .png files directly, rather than copying and pasting them into a work doc.**

**As for your minor and technical comments, we have implemented almost all of your suggestions.**

Minor comments:
1) Title: with "mixing" it is not entirely clear what you mean, would it make sense to point out here already the dry vs. moist convective mixing pathway of biomass burning aerosols (BBA) into the free troposphere (FT) that you are addressing in the paper?
   **Title changed to: "Detection of dilution due to turbulent mixing vs. precipitation**

**scavenging effects on biomass burning aerosol concentrations using stable water heavy isotope ratios during ORACLES"**

2) L 11-12: at this stage it is not clear to me that you mean dilution by turbulent mixing in the free troposphere. Maybe it would help me if you wrote "precipitation scavenging" and "dilution due to mixing". And in the next sentence, I would keep the same order of processes as in the first sentence: precipitation processes versus mixing.
**Sentenced changed as as suggested.**

3) L. 15: "Air… is distinct, and can be treated as separate analyses…" After reading the full paper I now understand what you mean but without knowing the paper this sentence sounds a bit mysterious. In which respect are FT and MBL air masses different?
**The abstract has been significantly reworded and restructured. This suggestion has been implemented as part of that process.**

4) L. 15: since you don't use MBL again in the abstract you don't need this abbreviation there, only in the main text.
**Removed.**

5) L. 16: remove "to"
**Removed.**

6) L. 17: "assess of…" I don't understand this sentence, something went wrong with the phrasing here. Maybe you mean "are used to assess the relevance of the air parcels' precipitation history"?
**Yes this was a phrasing mistake, fixed as suggested.**

7) L. 19: make clear that MERRA-2 is a global reanalysis dataset
**Changed to 'MERRA-2 global reanalysis dataset'.**

8) L. 28: CCN is not defined in the abstract
**Now defined.**

9) L. 31: Again make clear that you mean dilution by mixing: "signature of dilution effects by air mass mixing in the free troposphere". In my first reading I thought you mean dilution of heavy isotope concentrations by rain out underway. I didn't immediately think of mixing. But maybe making this clear at line 12 is enough. Of course, when reading your paper, one wonders what process exactly leads to dilution in the free troposphere. The turbulence is likely very limited except in the environment of clouds or in regions with strong vertical or horizontal wind shear…
**Changed as suggested**

10) L. 41: "particularly after again"? There seems to be part of the sentence missing here.
**Typo, should have read "particularly after aging", changed.**

11) L. 44: "…quantifying processes… is of importance for climate models."
**Changed.**

12) L. 54: maybe "compared to the reduction…"
**Changed.**

13) L. 54: this is the first time the term dilution comes up in the main text, make clear that you mean dilution by mixing through turbulence and convection.
**Added "(e.g. turbulent mixing during convection or cloud-top entrainment)"**

14) L. 57: how about first saying: "they provide information on the relative importance of the history of moist atmospheric processes such as mixing…"

**Changed as suggested.**

15) L. 60: "While isotope ratio information to date has shown promise…". This is true but a bit unspecific.
**This has been changed to "Previous studies leveraging heavy water isotope ratios have been able to partially but not fully constrain such processes. One reason for this is that…"**

16) L. 62: "… for the advantages to be realized". Maybe "to take full advantage of the tracer capacity of water isotopes and their use for assessing aerosol cycling in the atmosphere".
**Changed as recommended.**

17) L. 67: "The isotopic compositions for water D/H" is a pleonasm. Maybe "the isotope composition of water" or the "concentration of HDO"?
**Has been changed to "the dataset includes in-situ measurements of heavy water stable isotope ratios"**

18) L. 70-71: "In the case of the FT, regional aerosol and moisture transport are connected and characterized" what does "connected" mean exactly? That they are considered together?
**The paragraph was restructured, and in the process the statements was deleted.**

19) L. 82: How are the relative importance of evapotranspiration, dry versus moist convection etc. constrained? This needs to be explicitly clarified and the (q,dD) phase space is so essential for your paper that it needs to be introduced in the introduction.
**Revised. The concept of q, dD phase space is now introduced here.**

20) L. 69-90: this whole section is a bit unspectacularly written and sounds more like a casual report or list of things that were done in this paper (e.g. "Next,… is evaluated."), but in a way that is not rising the readers curiosity and attention. This paragraph is absolutely key and it is very cool! I think the authors could invest in building up some tension here, maybe listing the research questions, rising the reader's curiosity.

**This suggestion is appreciated. We decided to leave this section as is, it's not flashy but it works.**

21) L. 88: A new paragraph could be started at "Section 2 covers…".
**New paragraph added.**

22) L. 89: "Section 3 presents a brief **analysis** of the sub-cloud as a starting point for contrasting the **analysis** of the FT **analysis** of precipitation histories and scavenging". Please rephrase.
**Reworded to include "analysis" only once.**

23) L. 91: "precipitation metrics", not sure what you mean here, the ones derived in Section 3?

**Changed to "test the degree to which the isotopic signals can detect recent precipitation (e.g. before the signal is removed by subsequent boundary layer mixing)".**

24) L. 106: "Flights were typically every 2-3 days, which 7-9 hours in duration and…".

**Changed.**

25) L. 114: I would leave away "Measurements of" and just write "Potential temperature…".
**Changed.**

26) L. 146: "Using vertical profiles…" I don't fully understand this sentence. In particular,

why there is a logical link between "using RH over liquid water as a flag for the CL" and "profiles through CLs with broken clouds".

**We feel this statement is clear. For example, if the aircraft conducted a vertical profile but happened to pass through a gap in the cloud deck when reaching the relevant altitudes, lwc=0 even though we passed through the vertical region where the cloud deck resided.**

27) Section 2.5 first introduce the (q, dD) diagram, it is otherwise a bit difficult to follow and understand what is meant here.

**Moved section 2.5 to an appendix and is a highly technical note not needed to understand the rest of the paper.**

28) L. 171: What do you mean by "subsequent mixtures between them"?

**The first sentence has been changed to clarify, now reads: "For the FT analysis, it was desired to highlight hydrologic histories of distinct airmasses from the African continent (e.g. different BBA plumes), rather than any subsequent mixing between them as they advect westward to the SEA. Therefore, dDistinct FT airmasses in each vertical profile were identified using a pseudo-conservative variable approach"**

29) L. 187: There are no theoretical lines in Fig. 3, you probably mean Fig. 4.

**Yes Figure 4 should have been referenced. Changed.**

30) Fig. 3: is there a colormap missing or is it just that you color the points in the total water mixing ratio?

**Water mixing ratio is colored by altitude, so the vertical access in subplot (a) serves as a colormap. "colored by altitude" is now specified in the figure caption.**

31) L. 201: Rephrase, "for air where moisture from the respective evaporation source..." there is a verb missing or "and" is not needed.

**"and" deleted.**

32) L. 203: you probably mean the blue circle in Fig. 4.

**Correct, changed to Fig. 4.**

33) L. 208-220: In this paragraph one can get lost as an isotope beginner. Guide the reader, what is the aim? Also, it is not entirely clear which variable is measured which is modelled/taken from the reanalysis and what the uncertainty resulting from the use of the MERRA data is.

**An extra paragraph has been added which explains the aim of the convective detraining plume model in our study. Then, it is clarified that the algorithm using the MERRA data is to validate our use of the detraining plume model.**

34) Section 3: the start of the results section is a bit abrupt with a detailed description of Figure 5. The reader could be led a bit more smoothly through the results and its structure. The different results sections come a bit as a surprise.

**We agree upon review that the first results subsection is abrupt and unexpected. The primary reason for it is to validate our MBL measurements in (q, dD) space. E.g. show that their placement, which drives much of the subsequent analyses and assumptions, is founded in theory. However, it is not essential to the main results. It has been placed in an appendix and referenced in the results section, along with the above explanation.**

35) L. 227: Rephrase the first sentence, something went wrong. Maybe remove "of"?

**Removed.**

36) L. 230: nice that you refer to previous studies, but this could be done in the intro or the discussion, start with addressing your question. I think Feng et al. 2019

didn't present new observations.

**This is kept here since the section was moved to an appendix. Also, I don't believe we assert that Feng et al. present new observations.**

37) Results: From a structural point of view, and given your research questions sketched in the introduction, I am not convinced that starting with the subcloud layer is a good idea. But maybe there is a good reason for this, then state it to make sure the reader knows why he/she is reading this.

**Agreed, this section has been move to an appendix. See response to (34).**

38) Results and discussion: Given the complexity of this study, I very much doubt the use of separating the results and discussion section. Now that I made the effort of understanding your very interesting Fig. 5, I would like to learn what it tells me about moist processes in the boundary layer. Until Section 5 (where you discuss the science), I will have forgotten most of what you show in Fig. 5 because I will have had 10 additional complex figures to understand.

**Rather than restructure the results and discussion, section 3.1 has been moved to an appendix, keeping the paper's focus on the main analysis. Not having to address section 3.1 in the discussion also streamlines that section. We feel that this makes both the results and discussion sections focused and easy enough to follow (if detailed, we acknowledge) to leave them as separate sections.**

39) Fig. 5: When I look at Fig. 5, I don't really see these measurements as fitting particularly well in a mixing framework or along an imaginary mixing line.

**Nor are they meant to! The point of Fig. 5 is primarily to place the ORACLES measurements in some theoretical context, showing loose (if far from perfect) agreement.**

40) L252: what is the output frequency of the interpolated air parcel positions? Are hourly 3D wind fields used? The information on the trajectory calculation setup belongs into the methods section.

**As described in that section, the setup uses Global Data Assimilation System output on pressure levels at 1˚ x 1˚ resolution and 3-hour frequency. The output is at 1 hour frequency and has been added to the text. Since the description of HYSPLIT is only one sentence, we do not feel it needs to be relocated, as finding a place for it in the methods section would require a lot of restructuring to accommodate a single sentence.**

41) L258: "double that of the SCL" I don't' understand this. What is the CO concentration of the SCL, is it shown somewhere?

**This sentence has been revised for clarification, now reading: "It is noteworthy that a large fraction of the "low-BBA" air in the FTcan hadve CO a concentrations double that typically measured in theof the SCL during the campaign (not shown)."**

42) L262: remove it, "The low-BBA air in 2016 is…"
**Removed.**

43) L285ff: Make clear that this assessment with (monthly?) surface precipitation collocated with trajectory information is a first coarse assessment of the possibility of precipitation formation underway. It is not clear if 1) precipitation did really form at the time of the airmass overpass over a given region (timing uncertainty) and 2) if the airmass that you are tracking is really the one involved in precipitation formation, it could be that it is travelling above the cloud (vertical level of precipitation formation uncertainty)
**Comments added.**

44) L300ff: The sources of water vapour and BBA are likely not the same… This should be

critically discussed somewhere. Did the authors look at forest fire observations from remote sensing to identify potential BBA sources?

**Other studies, including those which supported the proposal for ORACLES, look at fire observations and sources for the BBA. We feel it does not need to be discussed here.**

45) L347ff: I am not sure, I understand why correlations are computed between dD and altitude, and in which context to place the vertical gradients of dD. Here the reader needs help to understand why these analyses are done.

**This is clarified now by adding the sentence: "If Rayleigh distillation followed by detrainment into dry air is in fact the primary process for the 2017 air parcels, one expectation is that $\delta D$ and altitude would have good correlation."**

46) Table 3: Was isoCAM evaluated with respect to near-surface observations over tropical Africa? How confident are the authors that this data is reliable? How is the

treatment of ET done in isoCAM? Is there a multilayer soil model to take near-surface soil layer enrichment due to evaporation? Is soil evaporation and plant transpiration treated in a fractionating or non-fractionating way?

**This is described in the first paragraph of Section 3.3.**

47) L385: why was this data discarded? It would provide a very interesting control group and reference point in the dD-q phase space for "clean" airmasses with likely important moisture contributions from the MBL.

**These are discarded because they likely did not undergo the convective detrainment model assumed for the analysis.**

48) Figs. 9 and 10 are complex and thus a bit too small to understand well.

**They were resized for the preprint and will be larger in the final version.**

49) In Fig. 11 many data points seem to be overlying each other. This makes the figure difficult to read.

**Scatter point size reduced.**

50) Fig. 11 and 12 and related results: I really like this approach, but I got lost when trying to understand how the convective plume model was exactly set up for these calculations. There are several unconstrained parameters in this model. I think a dedicated methods section on the setup of the convective plume model would really help me to see through the approach.

**The approximation to the model is described in L.411-417.**

51) To me turning back to the SCL and Fig. 5 now feels like quite a slalom.

**Not an issue now, since that analysis along with Fig. 5 are now in an appendix.**

52) L. 449: CCNs are already introduced as an abbreviation, define it above when first mentioning them in the main text.

**Changed.**

53) L496: As already mentioned earlier I find this part about the subcloud layer a bit weak. The motivation to assess the dD "for agreement with basic theory" is not very convincing. I like the idea of assessing the horizontal and vertical variability of the dD in the MBL and it would be nice to have 1 or 2 short statements about where this variability comes from process-wise.

**"Agreement" in this study just means "enough agreement to show that the isotope measurements have some basis in theory". Just enough to proceed more confidently with the subsequent analyses.**

54) L520ff: I am confused by the reasoning behind the first paragraph of this subsection: why should the high-BBA PDF be linked to the SCL observations? I thought that the hypothesis is that the transport pathway of moisture into the free troposphere is: 1) ET over tropical Africa, 2) dry or moist convective detrainment into the free troposphere, 3) Advection over the South Atlantic. So, if there is a link with the SCL measurements in the MBL that would be by convective downdrafts bringing dry FT air into the SCL, right?

**Yes your understanding of the hypothesis is correct. In the case of 2016, the hypothesis is that there is minimal ET contribution, which is why the SCL air works so well as an endmember. This is clarified now by adding the sentence: "This is interpreted as evidence of minimal ET contribution to the African PBL air, and therefore its moisture content closely resembles the original marine source which advected onto the continent."**

55) L558: I can't really see that in Fig. 13, to me it seems that many data points in all three IOPs deviate a lot from the mixing and Rayleigh lines.

**It is not how much they deviate from the mixing/Rayleigh lines that matters here, but how much the values of q and dD deviate from the SCL source. E.g. in 2016 the change in q averages -0.5 g/kg and change in dD is -2 permil, while in the other two years they are ~2-3x that on average.**

56) L560ff: I must have missed it but is the MBL defined as the SCL and the CL in your study? Could that be included in the sketch in Fig. 1.
**Added in the sketch of Fig.1.**

57) L568: that's very interesting!

**Agreed and thank you!**

58) L611: Repeating the fact that three IOPs over three years in the late winter period August-October with xx number of flights and in total yy hours of measurements were involved would be great.
**This has been added as suggested.**

59) L612: in combination with?

**Changed.**

60) L613: "of moisture contributions from different processes along the atmospheric transport pathways of water vapour (e.g....)"
**Phrase added.**

61) L615: "accompanied by cloud formation and precipitation..."

**Phrase added.**

62) L625 & L626ff: "This diagnostic partitioning cannot be achieved with humidity measurements alone..." very nice result indeed!!

**Agreed and thank you!**

63) L655: It would be fantastic if the paper could end with a more general closure statement of why this combination of isotopes with aerosol data is useful and why research on this combination of observations should be further pursued in the future.
**Final statement added.**

Technical comments:
- The Figures are all very interesting, but the quality (resolution?) is a bit poor. It's a pity, if it stays like this because of their very valuable content.
  **The figures look high quality and crisp in the web viewer version of the PDF for the preprint here: https://egusphere.copernicus.org/preprints/2023/egusphere-2023-69/egusphere-2023-69.pdf . But once downloaded some of the figures are blurred. We will keep an eye on this, but are expecting it to resolve itself for the final submission, as we will be submitting the .png files directly, rather than copying and pasting them into a work doc.**
- There are many instances with excessive spaces, double .. or ,, or missing commas.
  **The paper has been given a proof read to remove these occurrences.**
- To make this complex paper (in terms of the number of aspects and processes

involved) easier to access to the readership, I would strongly recommend reducing the number of abbreviations. There are some that are clearly needed, but others are used only a few times and could be written out. For example, for the boundary layer there is SCMBL, MBL, PBL… do you really need all of them or could you just abbreviate BL and then write out the specific one you mean? Or for example

biomass burning (BB) you don't use so many times, you mainly use BBA, so could you write out "biomass burning" in the few BB instances? I think similar considerations for other abbreviations would be worthwhile to make the paper easier to read.

- **Abbreviation "BB" has been removed.**
- **Instances of SCMBL and PBL have been removed.**
- **Southeast atlantic (SEA) abbreviation removed.**

- Many abbreviations are not properly introduced in the text, they should be written out the first time they appear (PBL, MBL, LCLT, CCN in the abstract).
  - **MBL instances fixed**
  - **PBL instances fixed**
  - **LCLT is a typo and has been removed.**

- Find a consistent term to refer to the (q,dD) phase space, it is sometimes (q,dD) and at other times q, dD. Also, this is such an important tool for the paper, could it be mentioned already explicitly in the abstract?
  **All instances changed to (q, dD), and it is also mentioned in the abstract now.**

- All variable names in italics, e.g. $R_D$ (L. 113). RH is sometimes in italics sometimes not (e.g. L. 152 and L. 155). However D as a chemical notation for deuterium should not be in italics.
  **Italics applied to variables where appropriate.**

- Airmass is sometimes written in one word sometimes in two (air mass). The same for end member vs. endmember.
  **All instances of endmember and airmass have been split into 2 words.**

- References mentioned directly in the text like Fiorella et al. (2022) have no comma:
  E.g. not Fiorella et al., (2022) at L. 316.
  **Instances of this have been corrected.**

**Authors' response to Referee #2's review of "Detection of mixing and precipitation scavenging effects on biomass burning aerosols using total water heavy isotope ratios during ORACLES" by Henze et al.**

**Dear Referee,**

**Thank you for taking the time to review our manuscript. Find our responses in bold below each of your comments/suggestions**

The mixing and precipitation scavenging effects are difficult to be quantified in observations. The authors showed a promising way by analyzing the observations of water isotope ratios and humidity together. The authors successfully show the difference between the mixing and scavenging processes with the new campaign data in the southeast Atlantic and their
humidity-isotope pairs analyses. I believe the method is novel to the field of the interaction between clouds and aerosols.

I don't have major scientific comments for the manuscript, though I think the manuscript should be proofread again. Since the topic of this manuscript is the biomass burning aerosols, I think the manuscript should highlight them more in the text. There are a lot of texts in the manuscript describing the convection processes. The authors can link them to the concentration of BBA.

**The manuscript has been proofread and revised in a way which we feel keeps the focus more on the black carbon scavenging. This includes revising some paragraphs as well as removing extraneous sections (e.g. Section 2.6 and 3.1, along with the discussion paragraph pertaining to 3.1).**

Minor comments:

**We have taken almost all of your suggestions below.**

Line 16: remove"to"? Line 17: remove "of "
**This sentence has been removed in the new revision.**
Line 28: What does IOP mean?
**Acronym removed.**
Line 41: after again?
**Should have been "after aging". Changed.**
Lines 52-53: "A component of both ….". The sentence reads weird. Why is the component of BC lifetimes the aerosol-cloud interaction?
**Sentence changed to "BC lifetimes in the atmosphere involve its interaction with precipitation, while cloud lifetime effects involve aerosol modification of precipitation"**
Line 70: Double "." Line 71: Double ","
**Fixed**
Line 78: double "topped"
**Fixed**
Figure 1: It would be nice to put two subplots in one row.

**The subplots have been placed in a single row.**

Line 96: Missing period after Redemann et al., (2021).

**Added**

Line 98: What does LCLT mean?

**Typo, changed to "lower FT"**

Line 101: deck due to -> deck is due to

**The sentence is "This air settles over the southeast Atlantic cloud deck due to large-scale subsidence, where it may then entrain into the MBL" which seems OK.**

Line 101: "where it": what does where refer to? Subsidence or cloud deck? What does "it" refer to?

**Sentence changed "This air settles over the southeast Atlantic cloud deck due to large-scale subsidence, and that air may then entrain into the MBL"**

Line 107: missing spaces in "100m" and "7km"

**Spaces added.**

Figure 2: Please add (a)(b)(c) for each subplot. Please choose a different color scheme for the left panel. It's not easy to tell three different tracks.

**These changes have been implemented.**

Line 146: "Using vertical profiles of T… was chosen": Please check the grammar here.

**Sentence changed to "Vertical profiles of $T$, $\theta$, and $RH$ were chosen over liquid water content to flag the CL since"**

Line 147: "it would also…": What does it refer to?

**"it" replaced with "the algorithm"**

Line 175: qdD is very important for this paper. Could you elaborate why qdD is useful than theta_e and what does it mean? Is it q times dD or something else?

**It is q time dD, which is less important for the paper than the (q, dD) diagrams (although related). This is a fairly technical part of the paper (compared to the real estate it was given) and is actually not too consequential for the final results. It has therefore been moved to an appendix.**

Figure 3: What do two profiles in the middle of Figure 3(a) represent? What does the multi-color mean? Should it be accompanied with a color bar?

**The profiles are explained in the figure caption. The coloring is by altitude, so the vertical axis serves as a colorbar. This figure has been moved to an appendix.**

Figure 209: Where are the "grey" curves? In Figure 3?

**Typo, should have been Figure 4.**

Figure 228: What is the spatial resolution of this COBE dataset? What does COBE mean? References?

**COBE meaning and spatial resolution added.**

Line 257: "low-BBA are": double "are" in this sentence.

**Second "are" removed.**

Figure 6: What is the difference between the dashed and solid line boxed regions? Also, no colors are filled in the color bar.

**The difference is explained in the figure caption and main text. There are color lines in the colorbar.**

Line 303: What's the resolution of this MODIS data? Please briefly introduce this dataset, including its duration.

**Resolution and duration of MODIS data included.**

Line 304: Which boxed region?

**Clarified that they are the regions bound by black/dashed squares**

Line 311: Table 3.2 or Table 2?

**Table 2, fixed.**

Line 317: 2022-> 2021

**Changed**

Line 323: figures -> Figures

**Changed**

Table 2: The row of "ocean evaporation source", the column of "calculation": equations 6-> Equation 6. "Horita and Wesolowski, 1994": Is it the correct citation style?

**Not sure what is being asked here, changed to Horita and Wesolowski, (1994).**

**Added parentheses.**

A weird "345" appears in the last row and last column.

**Something to do with Microsoft Word's line numbering. Fixed.**

Line 361: blue shaded region: light blue?

**Yes light blue, clarified in the caption now.**

Line 363: Table 3.2 -> Table 2

**Changed.**

Line 384: What does rBC mean?

**rBC stands for refractory black carbon (defined in the methods section). rBC is the measurement vs. BC is the quantity. However, line 384 has been changed to BC since the quantity is being referred to here.**

Line 415: Two "is" in one sentence.

**Extra is removed.**

Line 431: Please reduce the dot size to increase the visibility, or change this figure to a heatmap.

**Dot size has been reduced.**

Line 456: figures -> Figures

**Changed**

Line 501: add "the" between "advance" and "theory".

**That paragraph has been removed so no need.**

Line 534: close 0-> close to 0?

**Changed.**

Line 541: efficiency less than unity -> efficiency is less than unity

**Sentence changed to: "Possible reasons why this slope is less than 1 include variation in surface conditions during convection and precipitation efficiencies less than unity. "**

Line 575: "However": Two however in one paragraph.

**Second "However" removed.**

Line 584: I don't get what "either" means here.

**That sentence has been removed.**